# Sensitivity of Age of Air Trends on the derivation method for non-linear increasing inert $SF_6$

Frauke Fritsch[1,2], Hella Garny[1,2], Andreas Engel[3], Harald Bönisch[4], and Roland Eichinger[1,2]

[1]Deutsches Zentrum für Luft- und Raumfahrt (DLR), Institut für Physik der Atmosphäre, Oberpfaffenhofen, Germany
[2]Ludwig Maximilians University of Munich, Meteorological Institute Munich, Munich, Germany
[3]Institute for Atmospheric and Environmental Sciences, Goethe University Frankfurt am Main, Frankfurt am Main, Germany
[4]Institute of Meteorology and Climate Research, Karlsruhe Institute of Technology (KIT), Karlsruhe, Germany

**Correspondence:** Frauke Fritsch (frauke.fritsch@dlr.de)

**Abstract.** Mean age of air (AoA) is a diagnostic of transport along the stratospheric Brewer-Dobson circulation. While models consistently show negative trends, long-term time series (1975-2016) of AoA derived from observations show non-significant positive trends in mean AoA in the northern hemisphere. This discrepancy between observed and modeled mean AoA trends is still not resolved. There are uncertainties and assumptions required when deriving AoA from trace gas observations. At the same time, AoA from climate models is subject to uncertainties, too.

In this paper, we focus on the uncertainties due to the parameter selection in the method that is used to derive mean AoA from $SF_6$ measurements in Engel et al. (2009) and Engel et al. (2017). To correct for the non-linear increase in $SF_6$ concentrations, a quadratic fit to the time-series at the reference location, i.e. the tropical surface, is used. For this derivation, the width of the AoA distribution (age spectrum) has to be assumed. In addition, to choose the number of years the quadratic fit is performed for, the fraction of the age spectrum to be considered has to be assumed. Even though the uncertainty range due to all different aspects has already been taken into account for the total errors on the AoA values, the systematic influence of the parameter selection on AoA trends is described for the first time in the present study.

For this, we use the climate model EMAC (ECHAM MESSy Atmospheric Chemistry) as a test bed, where AoA derived from a linear tracer is available as a reference and modeled age spectra exist to diagnose the actual spatial age spectra widths. The comparison of mean AoA from the linear tracer with mean AoA from a $SF_6$ tracer shows systematic deviations specifically in the trends due to the selection of the parameters. However for an appropriate parameter selection, good agreement for both mean AoA and its trend can be found with deviations of about 1% in mean AoA and 12% in AoA trend.

In addition, a method to derive mean AoA is evaluated that applies a convolution to the reference time series. The resulting mean AoA and its trend only depend on an assumption about the ratio of moments. Also in that case, it is found that the larger the ratio of moments, the more the AoA trend gravitates towards the negative. The linear tracer and $SF_6$ AoA is found to agree within 0.3% in the mean and 6% in the trend.

The different methods and parameter selections were then applied to the balloon borne $SF_6$ and $CO_2$ observations. We found the same systematic changes in mean AoA trend dependent on the specific selection. When applying a parameter choice that is suggested by the model results, the AoA trend is reduced from $0.15\,\text{years}\,\text{decade}^{-1}$ to $0.07\,\text{years}\,\text{decade}^{-1}$. It illustrates that

correctly constraining those parameters is crucial for correct mean AoA and trend estimates and still remains a challenge in the real atmosphere.

# 1 Introduction

The Brewer-Dobson circulation (BDC) is the slow, overturning equator-to-pole mass circulation in the stratopshere (Butchart, 2014). As such it influences how chemical species are distributed in the atmosphere. For example, in the case of ozone depleting substances it is crucial to understand their transport to high latitudes, where they most severely impact ozone. As the BDC cannot be measured directly, age of air (AoA) has been established as a common measure to quantify it. Trace gas measurements can be used to derive AoA (Hall and Plumb, 1994; Volk et al., 1997). Both the residual circulation transport and mixing contribute to AoA (Garny et al., 2014). The quatitative relationship between AoA and the circulation were first established in pressure coordinates using the tropical leaky pipe model in Neu and Plumb (1999). Ray et al. (2016) improved upon this framework, incorporating other tracer information as well. Linz et al. (2016) isolated the diabatic circulation as a function of the horizontal difference in AoA between the tropics and the extratropics based on Neu and Plumb (1999). In a follow up paper, Linz et al. (2017) quantitatively calculated the overturning strength from tracer observations to compare with a model. From a conceptual point of view, AoA can be illustrated as a number of parcels that constitute an air mass and each of those parcels takes a different pathway from one reference point to an observation point. This idea was first introduced by Kida (1983). Those different pathways lead to a distribution of transport times, which is called the age spectrum of the specific air mass. The first moment of this age spectrum is the mean AoA.

Analyzing global chemistry-climate models very consistently shows a speeding up of the BDC in the case of climate change and negative trends in AoA (Oberländer et al., 2013; Shepherd and McLandress, 2011; Garcia and Randel, 2008). In contrast, both satellite and balloon borne observations show predominantly positive trends in AoA, (Engel et al., 2017; Stiller et al., 2012; Haenel et al., 2015; Bönisch et al., 2011). Predominantly as, e.g. Haenel et al. (2015) find a negative AoA trend in the southern hemisphere, Both observations and models have uncertainties in describing AoA. On the one hand, considering the anticipated dynamical variability (Hardiman et al., 2017), the balloon borne measurements are relatively sparse and the satellite measurements short (ten years) which limits the detection of forced trends. In the past, many model studies on BDC changes considered only residual circulation changes in pressure coordinates (e.g., Butchart et al., 2010), and it was shown that the residual circulation changes can partly be explained by the expansion of the atmospheric circulation (Singh and O'Gorman, 2012), i.e. the residual circulation trends at least partly disappear when considered relative to the tropopause height (Oberländer-Hayn et al., 2016). However, the strengthening of the BDC also holds in models when considering all transport processes, as measured by mean AoA (e.g., Eichinger et al., 2019).

Here, we want to focus on the uncertainties that arise from assumptions that are required to derive AoA from measurements. Those assumptions are necessary, as all tracers that are approximated as inert and measured in the atmosphere increase non-linearly. In our work, we will focus on $SF_6$ as a tracer. We will consider two methods to derive mean AoA here. First, the quadratic fit method was applied in Engel et al. (2009, 2017) which is highly relevant work regarding AoA trends. This

method considers the slope of the reference time series, which will be more closely explained in Sec. 2.3.1. The different contributions of uncertainty were considered very thoroughly for the total uncertainty of mean AoA in Engel et al. (2009, 2017). Another option to treat the non-linearity of the reference time series is the convolution with a specific Green's function (i.e. age spectrum) to deduce mean age, as for example used in Ray et al. (2017). This also allows the employment of more detailed approaches to the Green's function as shown in Andrews et al. (2001) and Bönisch et al. (2009). Leedham Elvidge et al. (2018) provides great insight to the total errors of mean AoA, applying both methods on measurements of various trace gases.

The latter also discusses the stratospheric lifetime of $SF_6$ and its effect on mean AoA. Both Leedham Elvidge et al. (2018) and Ray et al. (2017) find the lifetime of $SF_6$ to be lower than described so far. Up in the mesosphere $SF_6$ is depleted by photolysis and electron attachment (Totterdill et al., 2015; Kovács et al., 2017). The depleted air moves into the polar vortex and into the whole stratosphere as the vortex breaks down, leading to AoA biases varying across the atmosphere. The effect on mean AoA in stratospheric mid-latitudes is estimated to be around 0.75 years. In the present study, the focus is on the uncertainty due to the methodology of deriving mean AoA, therefore a $SF_6$ tracer without sinks is used.

In addition, Garcia et al. (2011) and Ray et al. (2014) worked on confining the uncertainty of mean AoA derived from $SF_6$ and $CO_2$ observations. Garcia et al. (2011) investigated AoA and its trends from a linear tracer and a $SF_6$ tracer with sinks in WACCM (Whole Atmosphere Community Climate Model). They corrected the non-linearity of the $SF_6$ tracer boundary condition (and the $CO_2$ tracer boundary condition) with an exponential growth correction. They find negative trends for AoA from the linear and non-linear tracers, with the non-linear being smaller. Ray et al. (2014) narrowed down the error bar of the observed trends by including additional measurements and correcting for differences in measurement locations. Even with those corrections, the trend remains of similar magnitude as calculated by Engel et al. (2009), albeit with smaller error bars.

Still, the systematic uncertainty of the trend of mean AoA due to every single assumptions of the specific methods are not yet described. Investigating the method in a chemistry-climate model allows the use of diagnostics that are not available for the real atmosphere and thereby to better understand the uncertainties due to the particular assumptions.

In Sec. 2 we will explain the methods under discussion, the trend calculation and provide details on the analyzed model simulations. We will present modeled age spectra and an analysis of the sensitivity of the trends on the parameter choices for a non-linear increasing tracer ($SF_6$) in Sec. 3. Sec. 4 discusses the results and relates them to the balloon borne observations. In Sec. 5 we will conclude our results.

## 2  Methodology

### 2.1  Age spectrum

The full information on the distribution of transport times is given by the actual age spectrum. The age spectrum is the distribution of transport times of trace gas concentrations to the location of interest $x$. Thus, it is equated to the Green's function ($G$) that describes the propagation of the tracer concentration $\chi_0$ from the reference location to the observed location. This can

be expressed with the following equation.

$$\chi(\boldsymbol{x},t) = \int_0^\infty \chi_0(t-t')G(\boldsymbol{x},t')\mathrm{d}t' \tag{1}$$

In which case $t$ denotes the transit time. The first moment of the age spectrum is the mean AoA $\Gamma$. This is written as

$$\Gamma = \int_0^\infty t'G(\boldsymbol{x},t')\mathrm{d}t'. \tag{2}$$

5  Consequently, the second moment of $G(t)$ is defined as

$$\Delta^2 = 0.5 \int_0^\infty (t'-\Gamma)^2 G(\boldsymbol{x},t')\mathrm{d}t'. \tag{3}$$

A well known approximation for $G(t)$ is given by an inverse Gaussian by Hall and Plumb (1994), which uses the first and second moment $\Gamma$ and $\Delta$

$$G(t) = \sqrt{\frac{\Gamma^3}{4\pi\Delta^2 t^3}} \cdot \exp\left(-\frac{\Gamma(t-\Gamma)^2}{4\Delta^2 t}\right). \tag{4}$$

10  This equation was obtained from a one-dimensional diffusion analog. Usually, $\Delta^2$ is parameterized relative to $\Gamma$ as $\Delta^2 = C_1 \cdot \Gamma$. Fig. 1 illustrates the effect of different values of $\Delta^2/\Gamma$ on Eq. 4. In a conceptual sense, $\Delta^2$ can be expressed relative to $\Gamma$ as the

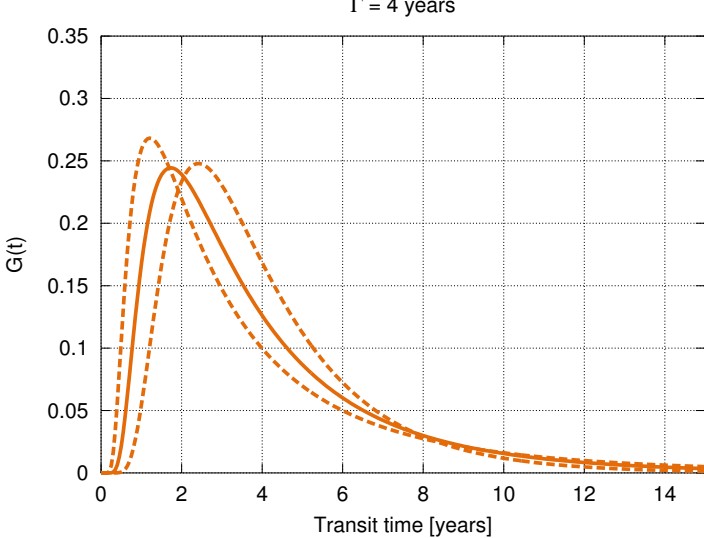

**Figure 1.** Sketch to illustrate the ratio of moments. The resulting age spectra from an inverse Gaussian with mean 4 years for values of $\Delta^2/\Gamma$ 0.7 years, 1.25 years and 2.0 years.

longer air has been traveling (i.e. larger $\Gamma$), the more spread out the age spectrum will be (i.e. larger $\Delta^2$). The specific value of $\Delta^2/\Gamma$ describes, how strong this spreading out is (Waugh, 2002).

More detailed approaches considered two-peak age spectra (Andrews et al., 2001). Though, in models the exact age spectrum can be obtained. If we consider Eq. 1, the Green's function can be obtained from a delta peak in tracer concentration. In the case of a non-stationary atmosphere, i.e. one with seasons, a set of inert tracer pulses at a reference surface is required. This method was already described and implemented for different models in e.g. Li et al. (2012) and Ploeger and Birner (2016).

## 2.2 Linear tracers

A well established way to derive $\Gamma$ in models is from a linearly increasing inert tracer. In this case, the mean age at a point $\boldsymbol{x}$ is equal to the time lag $\tau$ of the tracer concentration $\chi$ relative to the concentration $\chi_0$ at a selected reference area, which can be written as

$$\chi(\boldsymbol{x},t) = \chi_0(t - \Gamma(x)). \tag{5}$$

## 2.3 Non-linear tracers

### 2.3.1 Quadratic fit to the reference time series

In the real world, no perfectly linear increasing inert tracer is available. Often, $SF_6$ is used as a proxy for such a tracer. As the $SF_6$ emissions increase non-linearly, a common approach to derive mean AoA is a second order approximation that considers the slope of the reference time series. The general concept of the derivation is described in Volk et al. (1997). Engel et al. (2009) and Engel et al. (2017) apply it as an iterative procedure to optimize the length of the fit interval to the reference time series. First, the second order fit is performed on the reference time series for a set fit interval, e.g. 15 years. This yields the polynomial coefficients $a$, $b$ and $c$. The fit performed can be expressed either relative to the nominal time axis $\tilde{t}$ ranging from 1960 to 2011 and the reference time $t_0$ or by the transit time $t$ (note the change in sign):

$$\chi_0(\tilde{t}) = a + b(\tilde{t} - t_0) + c(\tilde{t} - t_0)^2 = a - bt + t^2 \tag{6}$$

Secondly, a first guess for the mean age is calculated using the coefficients applying the equation (Volk et al., 1997)

$$\Gamma = \frac{b}{2c} - C_1 \pm \sqrt{\left(\frac{b}{2c} - C_1\right)^2 + \tau^2 - \frac{b}{c} \cdot \tau}. \tag{7}$$

Again, $\Delta$ is the width of the age spectrum and $C_1$ is the ratio of first and second moments of the age spectrum omitting a factor of two, i.e. $C_1 = \Delta^2/\Gamma$. $C_1$ parameterizes the width of the spectrum relative to its mean. Again, $\tau$ is the lag time between the concentrations $\chi$ and $\chi_0$. To be consistent with Engel et al. (2017), the lag time is determined by approximating the reference time series by a second order polynomial. In the third step, a fit interval $t_{\text{fit}}$ is determined, which considers a given fraction of input ($F$) of the age spectrum, e.g. 98%. This can be written as:

$$\int_0^{t_{\text{fit}}} G(t')\mathrm{d}t' \stackrel{!}{=} F \tag{8}$$

More schematically, the fraction of input is illustrated in Fig. 2. Considering a given fraction $F$ of the area enclosed by $G(t)$ gives a transit time that corresponds to this fraction, and this transit time is then used as backward fit interval to perform the fit on the reference time series. Thus, the larger the fraction of input F, the longer the fit interval. The age spectrum $G(t)$ assumed

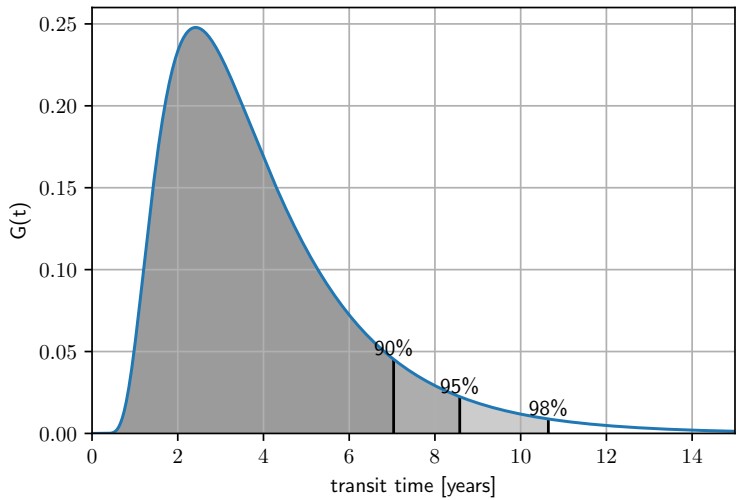

**Figure 2.** The resulting fit intervals for the fraction of input $F = 90\%$, $95\%$ and $98\%$ are sketched. $G(t)$ is assumed from an inverse Gaussian for $\Gamma = 4$ years and $\Delta^2 = 0.7 \cdot \Gamma$.

in this step is the inverse Gaussian from Eq. 4. This equation uses the previous mean age value and the parameterization for the
width. In the last step the value for the fit interval is used to determine once more the polynomial coefficients and recalculate the mean age value according to Eq. 7.

Summing up, to perform this procedure one needs to choose the reference point, the fraction of the considered input and the ratio of the moments. Our aim is to systematically test the sensitivity of the derived mean AoA and AoA trends on those parameters.

### 2.3.2   Convolution Method

Another method to infer mean AoA from a non-linear increasing tracer is using a numerical convolution. Considering Eq. 1, the reference time series $\chi_0$ is convoluted with an assumed age spectrum $G$, i.e. Eq. 4, which gives an observed mixing ratio $\chi$. Inserting different mean age values $\Gamma$ into Eq. 4, creates a look-up table of concentrations $\chi(\Gamma)$ that can be compared to the actual observed mixing ratio to pick the appropriate mean age value. This approach requires one less assumption than the
fit method as an assumption about the considered amount of reference input is not needed as the age spectrum weights the reference input according to the relative influence it has at the observed location. Therefore, the reference input can simply be considered for a long time interval, e.g. 25 years. Thus, this approach is expected to be quite physical and robust. Ray et al. (2017) and Leedham Elvidge et al. (2018) apply this approach to derive mean AoA from $SF_6$ and other non-linear increasing

tracers, respectively. The AoA calculations in Stiller et al. (2012) are also based on this concept, though they determine $\Gamma$ by an iterative process rather than by a lookup table.

## 2.4 EMAC Simulations

In the present study, we use the results of two simulations of the chemistry-climate model EMAC (ECHAM/MESSy Atmospheric Chemistry, Jöckel et al., 2010), which are both in a T42 horizontal (2.8 x 2.8 degrees) resolution with 90 layers in the vertical and explicitly resolved middle atmosphere dynamics (T42L90MA). In this setup, the uppermost model layer is centered at around 0.01 hPa and the vertical resolution in the upper troposphere lower stratosphere region (UTLS) is 500-600 m. MESSy (Modular Earth Submodel System) provides an infrastructure with generalized interfaces for standardized coupling of Earth System Model (ESM) components (dynamical cores, physical parameterizations, chemistry packages, diagnostics etc.). The dynamical core of EMAC is the European Center Hamburg general circulation model (ECHAM5, Roeckner et al., 2006).

The first simulation is a free running transient hindcast simulation using EMACv2.51 with full interactive chemistry stretching from 1960 to 2011. This simulation was performed within the ESCiMo project (Earth System Chemistry integrated Modeling, Jöckel et al., 2016) and is referred to as RC1-base-07. The setup of this simulation follows the REF-C1 scenario as defined by the Chemistry-Climate Model Initiative (CCMI, Eyring et al., 2013). Details on the setup and an evaluation of this chemistry-climate simulation can be found in Jöckel et al. (2016) and in Morgenstern et al. (2017). The simulation had ten years of spin-up, i.e. 1950-1959. This means, that for the earliest AoA values calculated for 1975 the respective fit back along the reference time series does not stretch into the spin-up period. Previous tests yielded that this spin-up length is sufficient to not affect mean AoA.

The second simulation, termed tranPul, is an EMACv2.53.0 run which is designed to resemble the RC1-base-07 simulation as closely as possible, however for cost efficiency reasons, without using interactive chemistry. This means that only the EMAC modules for dynamics, physics and diagnostics were used, namely AEROPT, CLOUD, CLOUDOPT, CONVECT, CVTRANS, E5VDIFF, ORBIT, OROGW, PTRAC, RAD, SURFACE, TNUDGE, TROPOP, VAXTRA. For details on these submodels refer to Jöckel et al. (2005, 2010, 2016). In that simulation, we transiently prescribed the monthly mean fields of the main greenhouse gases $CO_2$, $CH_4$, $N_2O$ and $O_3$ from the RC1-base-07 simulation as zonal mean, monthly mean fields. As both simulations do not include a coupled ocean, SSTs and SICs have been prescribed following the global data set HadISST provided by the UK Met Office Hadley Centre (available via http://www.metoffice.gov.uk/hadobs/hadisst/, Rayner et al., 2003). Although in the L90 setup of EMAC, the quasi-biennial-oscillation (QBO) is internally generated (Giorgetta et al., 2002), the zonal winds near the equator were slightly nudged in the simulations to get the correct phasing of the observed QBO. Moreover, atmospheric aerosol (including volcanic aerosol) has been prescribed to take into account the interactions with radiation (in both simulations) and with heterogeneous chemistry (in the RC1-base-07 simulation). Again, ten years of spin-up were considered.

The RC1-base-07 simulation is used to study the methods that are used to derive AoA from non-linear tracers, such as $SF_6$. As our focus is to quantitatively assess the uncertainties of the mean AoA derivation, the SF6 tracers considered in these simulations do not include mesospheric sinks. The $SF_6$ time series prescribed in the simulation follows the RCP 6.0 scenario (Eyring et al., 2013). Its global mean is shown in Fig. 3 as it is prescribed in the simulations to emulate emissions. The $SF_6$

time series as applied with the calculations from measurements is also displayed (see Levin et al., 2010; Maiss and Levin, 1994). From 1978 on, the time series is well constrained by NOAA measurements at Cape Grim. Before that, $SF_6$ is estimated from emissions. Prior to 1961, the data is linearly extrapolated. Both $SF_6$ time series differ slightly, as can be seen from Fig. 3. However, as explained in detail in Sec. 3.2, this does not impact our method evaluation.

In the tranPul simulation, periodical pulses of an inert tracer were released at the surface in the range between $20°$ S and $20°$ N. This means that every three months, the lower boundary conditions of that tracer are set to 1 for one month at the tropical surface. After 9.5 years, the tracers are set to zero and started again to avoid large computational expenses. This operation is performed with the specifically designed MESSy submodule TPULSE. A similar EMAC setup was already applied in Hauck et al. (2019). The pulses allow the derivation of time evolving age spectra in EMAC as described for CLaMS (Chemical

Lagrangian Model of the Stratosphere) in Ploeger and Birner (2016).

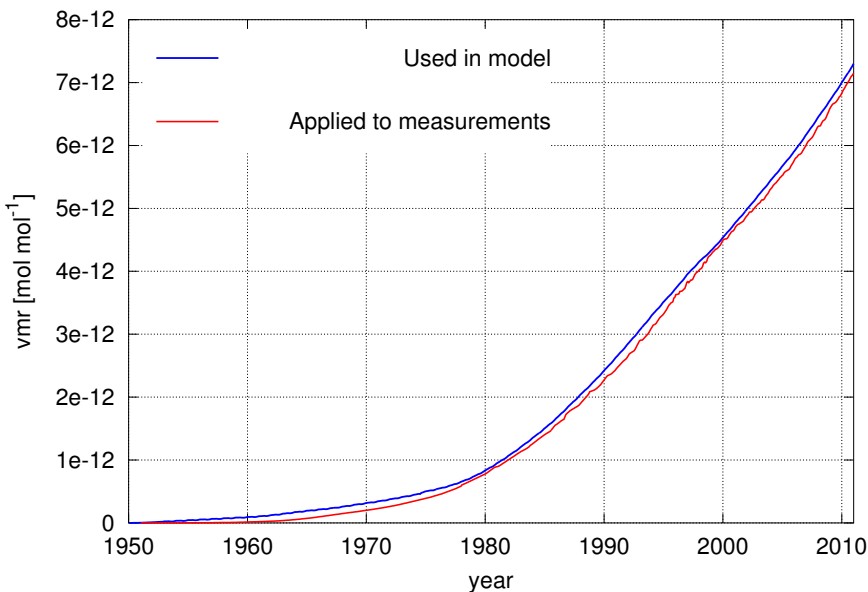

**Figure 3.** Globally averaged $SF_6$ time series on the ground as used in the RC1-base-07 simulation (blue) and as used with the observations calculations (red).

## 3  Results

### 3.1  Age spectra

It is straightforward to derive mean AoA from linearly increasing tracers in models, as described in Sec. 2.2. To give an example of mean AoA in EMAC, Fig. 4 shows mean AoA derived from the idealized linear increasing tracer for July 2001-

2010 from the RC1-base-07 simulation. Since the derivation of mean AoA from measurements requires an assumption about

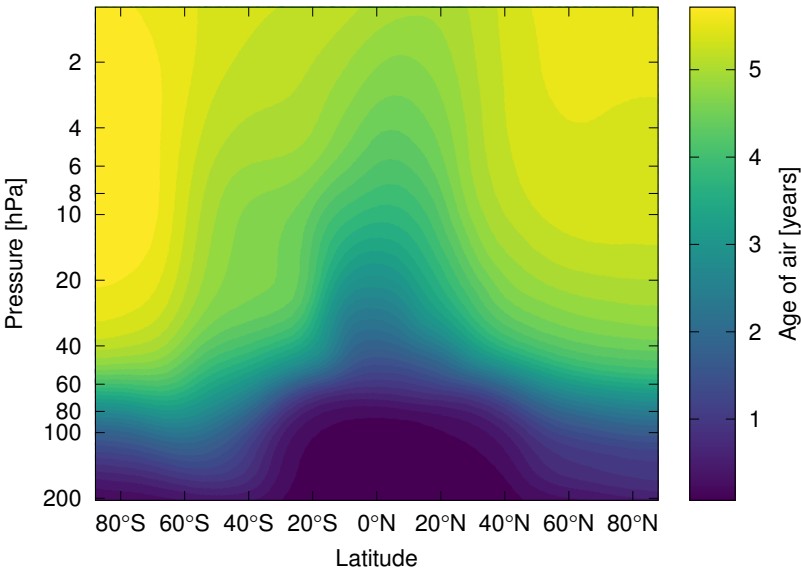

**Figure 4.** Mean AoA derived from the idealized tracer from the RC1-base-07 ESCiMo simulation. July mean for the years 2001-2010.

the AoA spectrum, we derived spectra for EMAC in a transient simulation. The implementation was such, that tracer pulses were released every three months and kept running for 9.5 years, which is due to the technical realization. Fig. 5 shows the seasonal evolution of the modeled spectra at $59°$ N and 106 hPa. This is close to 400 K and $60°$ N potential temperature, as in Figure 5 in the study by Ploeger and Birner (2016). The MAM, JJA and SON spectra show a similar propagation to Ploeger and Birner (2016). DJF rather shows one major peak instead of two as in Ploeger and Birner (2016). Our coarse resolution of the spectra along the transit time axis of 3 months could be the reason for this, since the actual maxima of the age spectrum may fall in between a three month period such that they are not visible. For comparison, the spectrum calculated from the Hall and Plumb parameterization (see Eq. 4) is shown, using the values from the annual mean spectrum, that is a mean age value of 1.6 years (vertical line) and a ratio of moments of 1.8 years. For the chosen parameters, the modal age of the theoretical Hall and Plumb spectrum is lower than for the actual spectrum and the tail is broader. However, how well the modal age compares depends strongly on the applied ratio of moments.

The ratio of moments calculated directly from the spectra as shown in Fig. 6 (left) are small (considering the length of the spectra of 9.5 years), considering values of 0.7 to 1.25 years are proposed for age calculations in Hall and Plumb (1994). Following Ploeger and Birner (2016), the spectra were extended to 50 years using an exponential fit on the second half of the spectrum, i.e. transit times above 5 years. The ratio of moments in that case (Fig. 6 right) shift to larger values of up to 1.7 years in the annual mean covering the full simulation. This is consistent with e.g. Hauck et al. (2019), who also find ratio of moments larger than the upper limit of 1.25 years derived from the one dimensional diffusion analog by Hall and Plumb (1994). Though, it should be considered that Hall and Plumb (1994) assumed 10 years of age spectrum in their approximation, while the other

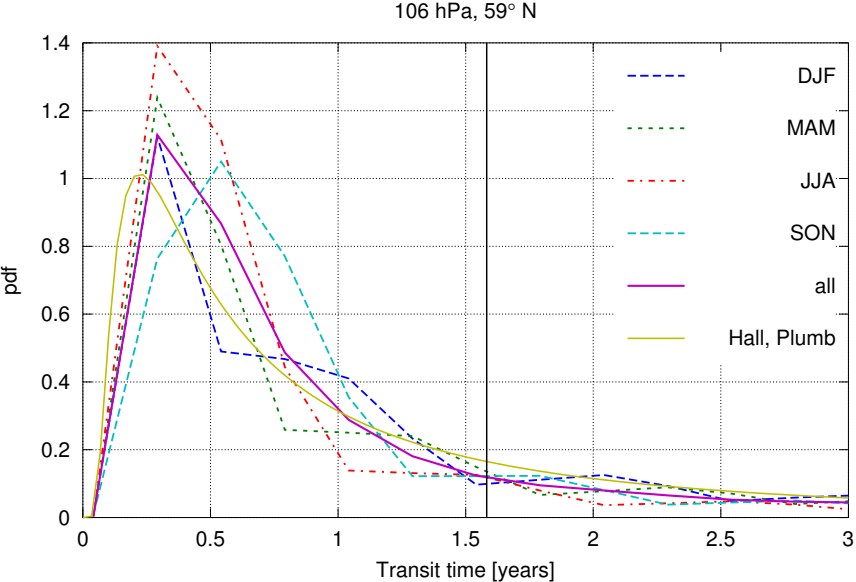

**Figure 5.** Age spectra at $59°$ N and $106$ hPa from the tranPul simulation. The dashed lines are the seasonal mean spectra. The pink solid line is the annual mean age spectrum. The solid yellow line is the spectrum calculated from the Hall and Plumb parameterization using the values from the integration of annual mean spectrum, that is a mean age value of 1.6 years (vertical line) and a ratio of moments of 1.8 years (Eq. 2 and 3)

approaches also considered longer transit times, which tends to give larger ratio of moments. More details on the interannual variability of the ratio of moments will be discussed in Sec. 3.2. Also in that section, we will use the ratio of moments from the modeled spectra in the age calculations.

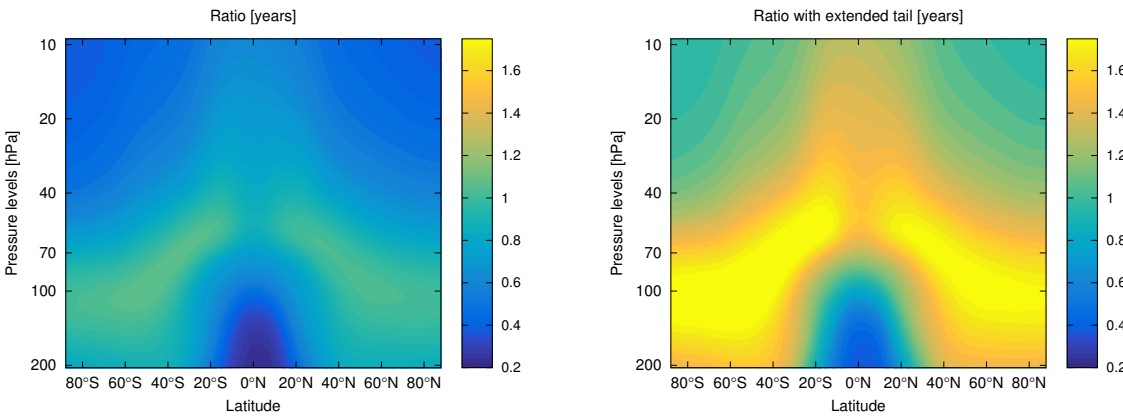

**Figure 6.** Mean ratio $\Delta^2/\Gamma$ derived from age spectra in the tranPul simulation. On the left for the regular 9.5 year long spectra, on the right with exponentially extended tail as in Ploeger and Birner (2016).

## 3.2 Sensitivity of mean AoA and AoA trends on parameter choice in the fit method

As described in Sec. 2.2, mean AoA and its trend were calculated from $SF_6$ and an ideal linear tracer for the RC1-base-07 simulation for a range of parameters and references for the fit method (explained in Sec. 2.3.1). Since we are considering a self-consistent model world and idealized $SF_6$ without sinks implemented, we expect perfect agreement between the trends derived from the two tracers.

The fraction of input was varied between 90%, 95% and 98% and the ratio of moments between 0.7, 0.8, 1.0 and 1.25 years. This covers the range for the ratio of moments proposed in Hall and Plumb (1994). Also, the pair of values 98% and 0.7 years, as applied to the measurements in Engel et al. (2009) and Engel et al. (2017), is considered in this selection.

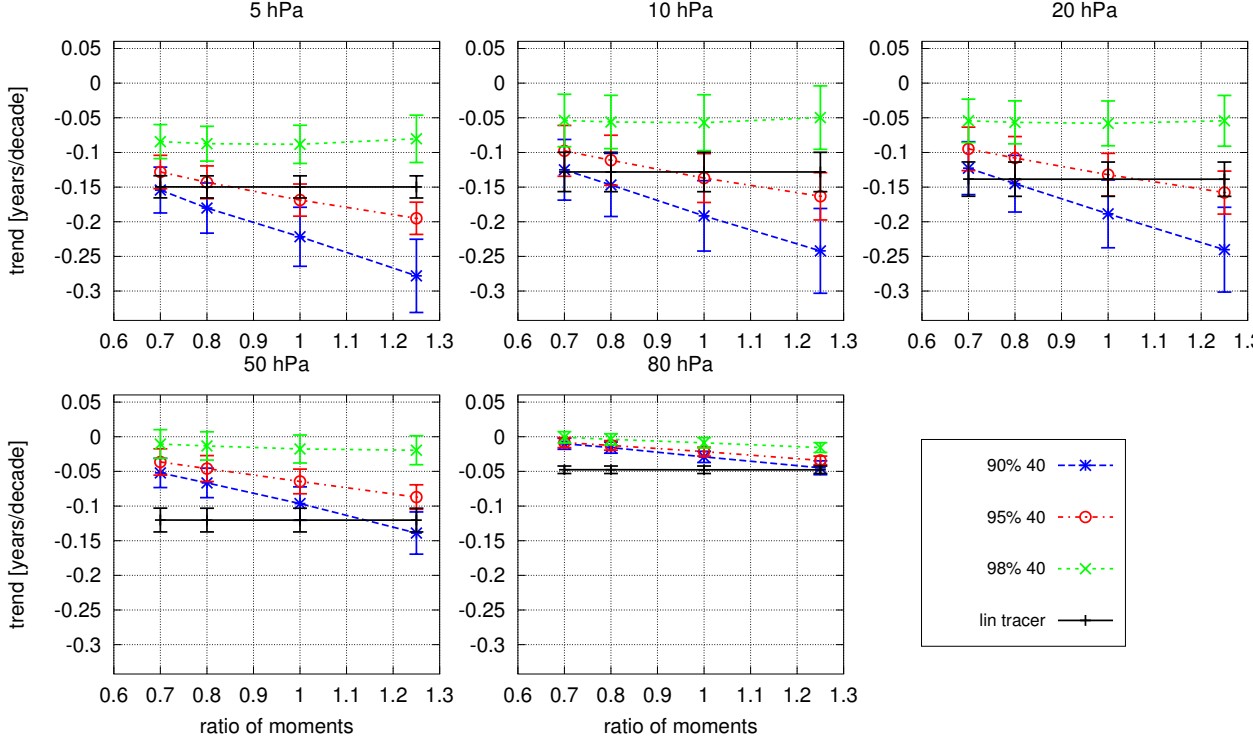

**Figure 7.** Overview of the trend in mean AoA derived from $SF_6$ (dotted lines) and a linear tracer (solid line) for different ratio of moments (x-Axis) and fractions of input (colors) at 40° N. The panels show different pressure levels. The trend calculation considered the time series from 1975 to 2011. The error bars show $1\sigma$ which is the 68% confidence levels.

The resulting trends for 40° N and a selection of pressure levels are shown in Fig. 7. Other latitudes show similar behavior and are therefore not shown. The trend calculation was performed on deseasonalized values and considered autocorrelation (Thompson et al., 2015). The trend of the linear tracer is included in the plots for each parameter choice to make comparison easier, though it does not have any sensitivity to them. Overall, the trends derived from $SF_6$ are more negative the larger the ratio of moments and closer to zero the larger the fraction of input that is considered. Furthermore, the sensitivity of the trends

to the ratio of moments increases when using a lower fraction of input. Intuitively, one might think that the results would be better the more of the reference input is considered (larger fraction of input). However, the fit performed on the reference time series is not weighted by, e.g. the spectrum. Therefore taking very long fit intervals takes into account more old air, which only has a small contribution to the actual mean AoA.

Fig. 7 also shows that depending on the pressure level, different selections of ratio of moment and reference input lead to agreement between the trends of the idealized and the $SF_6$ tracer. The influence of transport and mixing on the age distribution can explain that the best choice for the ratio of moments varies by location.

To build onto what was already seen from Fig. 7, Fig. 8 shows the mean absolute deviation of mean AoA derived from $SF_6$ and the linear tracer in the latitude-height domain for different selections of parameters in the derivation (colors). The white contours show the difference in trends in $10^{-1}$ years decade$^{-1}$. Further, in Fig. 8 we also consider a spatial distribution of ratio of moments. Even though we do not have age spectra available for the RC1-base-07 simulation, those from the tranPul simulation are a suitable first estimate for EMAC as a model. We considered the ratio of moments for the 9.5 year long spectra (PulRa sh) and the extended spectra (PulRa) (see Fig. 6).

It is evident from Fig. 8 that using fixed ratio of moments for the whole latitude-height domain always leads to deviations in mean AoA and its trends in some region (Panels a-f). Using the ratio of moments derived from the 9.5 year spectra reduces the spatial pattern of deviations (panel g). However, an offset to the actual mean AoA values and trends remains as overall the ratios of moments are underestimated when neglecting the tail. In the range of the balloon borne observations (5 hPa-30 hPa and 32° N-51° N) this means an average deviation of 4% in AoA values and 56% in AoA trend. The ratio of moments derived from the extended age spectra leads to the best agreement across the widest spatial range for both mean AoA and its trend (Panel h). Both annual mean ratio of moments and transient ratio of moments, were used, though the transient ratio of moments do not show strong further improvement over the annual means (Panel i). Using PulRa and 95% fraction of input provides 1% average deviation in age values and 12% deviation in trends in the region of the observations. The spatial dependency of an appropriate ratio of moments might play a role for the measurement application considering that the measurements in Engel et al. (2009) were analyzed between 5 hPa-30 hPa and 32° N-51° N.

Fig. 9 shows an example of the transient ratio of moments at 20 hPa and 40° N. Its trend is drawn, even though it is not significant as the trend is small compared to the variability. The trend of the ratio of moments is negative at all locations, but mostly not significant. The cycles visible in the ratio of moments can origin from the prescribed QBO and the solar cycle inherent in the prescribed ozone. It is beyond the scope of the current study to examine the large variability in the ratio of moments, but in particular the strong variation with periodicity of about 10 years calls for future investigations. Considering for example Labitzke (2007) it is anticipated that the solar cycle influences the BDC and thereby the shape of the age spectrum. The AoA calculation typically considers about 10 years of reference input. That equates roughly to one cycle in strong variation in ratio of moments which are integrated. That might explain that most improvement in the AoA derivation comes from applying an appropriate spatial distribution of ratio of moments and not from using a transient one.

As shown in Fig. 8, Panel f, the combination of $F = 98\%$ and $\Delta^2/\Gamma = 0.7$ years applied to derive AoA from observations leads to systematic offsets in both mean AoA and the AoA trends between the linear and $SF_6$ tracer in EMAC. In Fig. 10

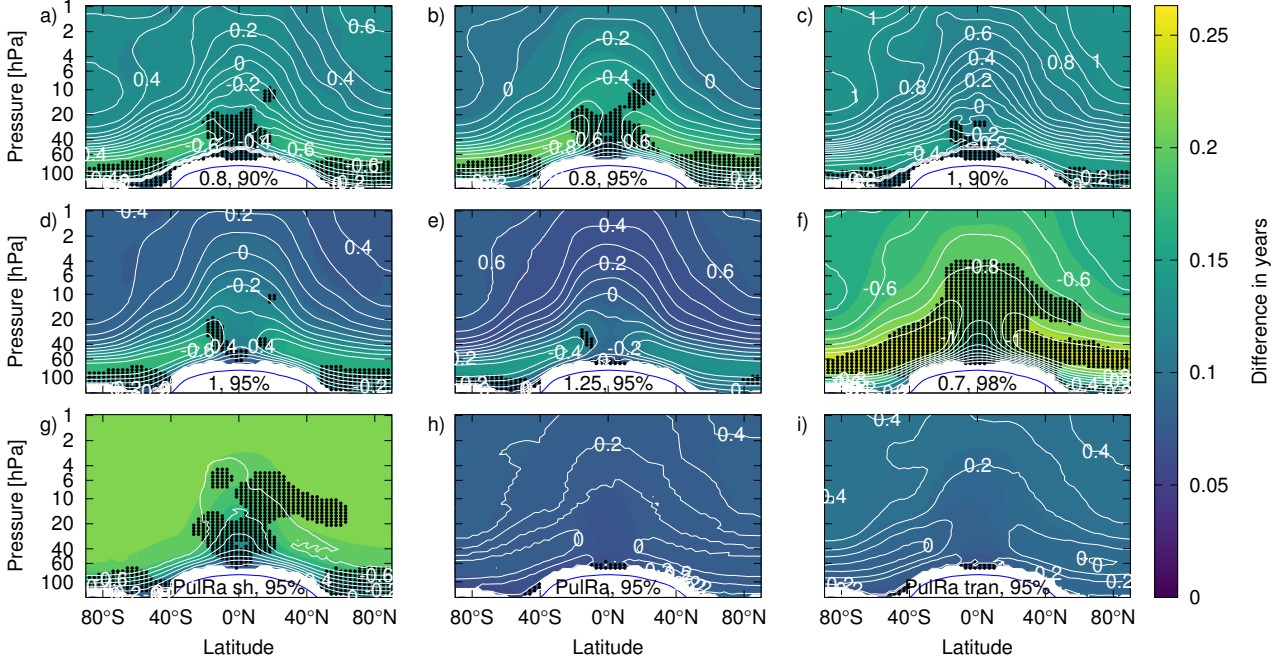

**Figure 8.** Mean absolute deviation of mean AoA calculated from SF6 from AoA derived from an ideal tracer for different ratio of moments and fraction of input (colors). The overlaid white contours are the difference in trends in $10^{-1}$ years decade$^{-1}$. At the bottom of each panel, the fraction of input and ratio of moments considered in the derivation are stated. PulRa denotes the ratio of moments derived from the tranPul simulation annual mean extended age spectra, PulRa sh is from the 9.5 year spectra. PulRa trans uses the transient extended spectra from that simulation. White areas indicate where the derivation is not applicable. Black stippling show insignificance of the SF6 trend from zero on the 95% confidence level. The trend calculation considered the time series from 1975 to 2011. The tropopause and the range 30 K above are excluded, as the AoA calculation is not defined in this region due to the different entry regions into the stratosphere that have to be considered.

the mean AoA trends are shown, which further illustrates the offset in trends. The AoA trends derived from SF$_6$ with this parameter choice are slightly negative or even positive and insignificant across a wide domain. Calculating the average AoA trend for 98% and 0.7 years across the spatial range of the observations, i.e. 5 hPa-30 hPa and 32° N-51° N, yields

- $-0.13$ years decade$^{-1}$ $\pm$ 0.025 years decade$^{-1}$ for the linear tracer

5 - $-0.057$ years decade$^{-1}$ $\pm$ 0.033 years decade$^{-1}$ for the SF$_6$ tracer.

Hence, averaged across that range, the mean AoA trend derived using those parameters stretches to positive on the $2\sigma$ range (95% confidence). Expressed relatively, these represent average deviations of 4% in AoA values and 58% in AoA trend.

Since the reference location for the AoA calculation can also be varied in model calculations, we checked the effect on the resulting AoA and its trends. However, the different reference locations that have been tested do not show a strong influence

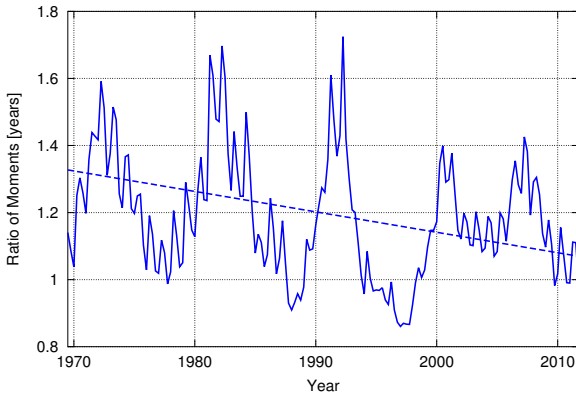

**Figure 9.** Ratio of moments derived from extended age spectra from the tranPul simulation at 20 hPa and 40° N

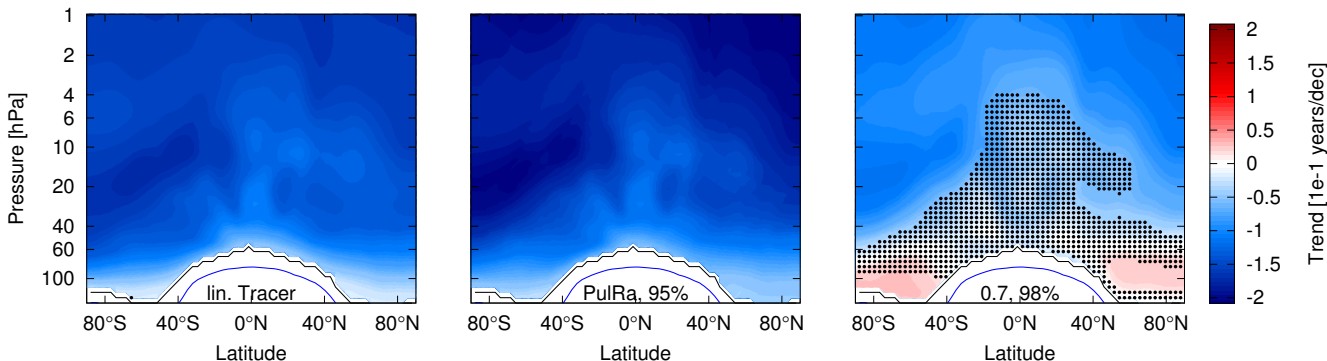

**Figure 10.** Trends of mean AoA from 1975 to 2011 derived from a linear tracer (left panel) and from $SF_6$, for the ratio of moments from the extended pulse spectra and 95% fraction of input (middle) and 0.7 years ratio of moments and 98% fraction of input. White areas are the areas that can not be analyzed for a trend and black stippling shows insignificance of the trend from zero on the 95% confidence level using a t-test.

on the resulting mean AoA or its trends. For example, an average of the concentration in the area between 20° S and 20° N on the ground and the same range at 100 hPa were used. In that case, the differences in mean AoA are less than half a year and represents the offset time that the air traveled between the different reference locations (not shown). The differences in the trends are below 0.01 years decade$^{-1}$. Both statements refer to the results we obtained for different selections of fraction of

5 input and ratio of moments, i.e. also considering selections that do not perform well overall. However, the reference location is important for quite specific tracer initialization on the ground, as for example if the linear AoA tracer is initialized from the poles. In such cases a reference location closer to the tropopause should be used to avoid artifacts of this tracer initialization in the AoA calculation. Since this is not the case for our simulations, we used the ground as reference in all calculations shown here as it is more applicable to measurements.

We find that the sensitivity of the AoA trends derived from $SF_6$ to the parameter choice arises predominantly from deviations in mean AoA in earlier years (1970s and 80s). As an example for this, the left of Fig. 11 shows the mean AoA time series and the derived trend at 20 hPa and $40°$ N for the combinations 98% and 0.7 years (red) and 95% and 1.25 years (blue). The linear tracer is given in black. 95% and 1.25 years represents a combination with relatively good agreement with the linear tracer in AoA whereas 98% and 0.7 years shows a strong deviation. For the mean AoA values at this location the length of the fit to the reference time series is around 11 years. The right panel of Fig. 11 shows the differences of the $SF_6$ derived mean AoA and the linear tracer derived mean AoA.

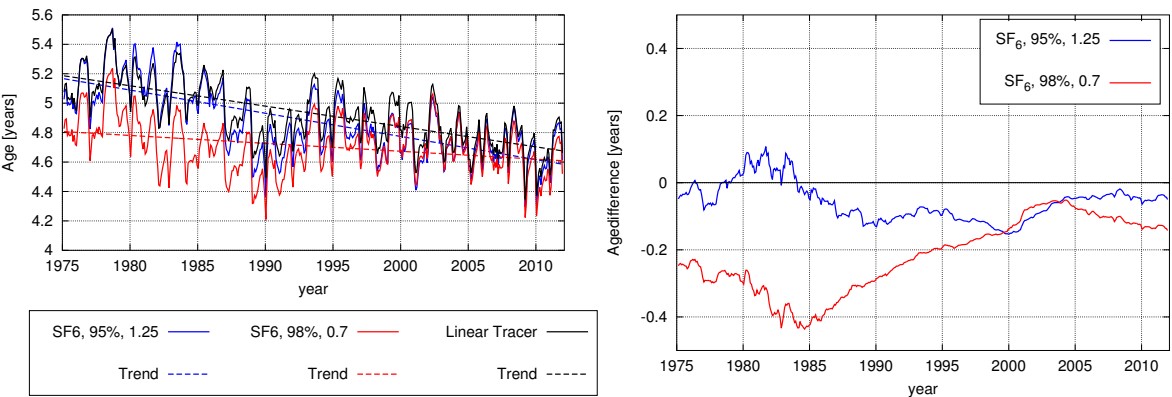

**Figure 11.** On the left mean AoA and its trend derived from $SF_6$ for two sets of parameters (red and blue) and from the linear tracer (black) at 20 hPa and $40°$ N from 1975 to 2011 from the RC1-base-07 Simulation. On the right accordingly mean AoA from the $SF_6$ tracer minus mean AoA from the linear tracer.

Considering Eq. 7, one explanation for the deviations might be that at the beginning of the reference time series the specific fit coefficients $c$ and $b$ of the second order polynomial are very sensitive to the fit interval length. Therefore Fig. 12 shows the coefficients dependent on the start date of the fit and the number of years fitted back. The parameters vary slightly stronger for fits early in the time series. As can be seen from Fig. 12 and already from the $SF_6$ time series in Fig. 3, the first part of the $SF_6$ time series is rather linear. In the eighties the second order contribution is strongest. Afterwards, the slope is rather linear again. Still, the times which show more variation in the coefficients with changing fit interval length do not clearly correspond to the times that the mean AoA derived from $SF_6$ deviates more from AoA derived from the linear tracer.

Another explanation for the variation in mean AoA might be the sensitivity of the lag time $\tau$ to the fit interval length. To be consistent with Engel et al. (2009) and Engel et al. (2017), the lag time is determined by approximating the reference time series as a second order polynomial function of the concentration. Thus depending on the number of years considered, the obtained lag time might vary. This is counterintuitive to what one would expect if the lag time was directly looked up from the concentrations. Fig. 13 shows the lag times found for the $SF_6$ time series for fit interval lengths from 9 to 15 years at the same location as the example in Fig. 11. From roughly 1983 to 1990, $\tau$ depends stronger on the length of the fit interval and varies up to 12% for the shown range of fit intervals.

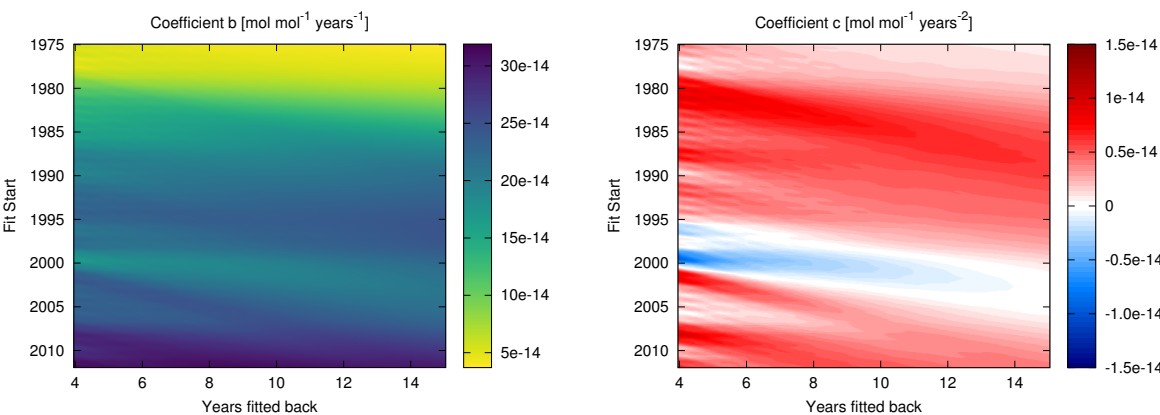

**Figure 12.** Coefficients $b$ and $c$ of the second order polynomial fit to the $SF_6$ reference time series dependent on the chosen length of fit interval.

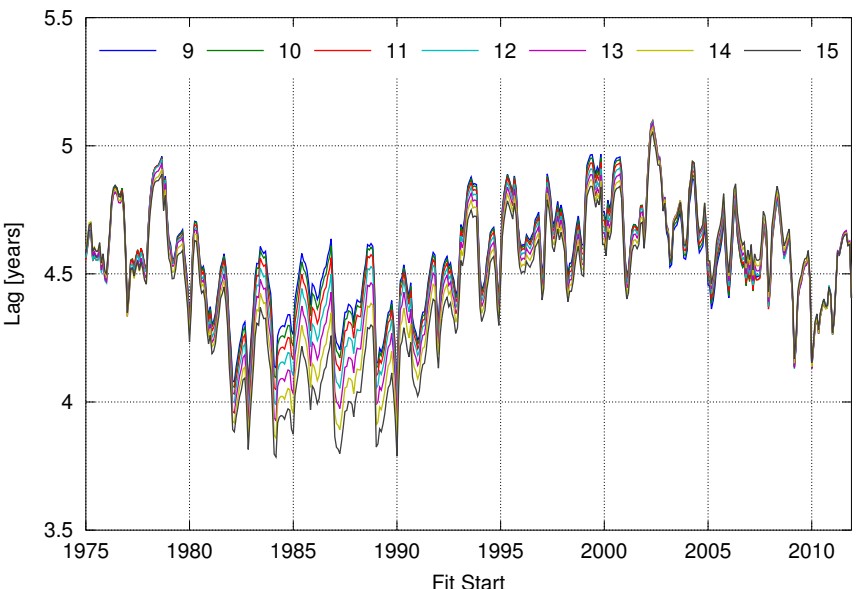

**Figure 13.** Lag time $\tau$ for fit interval length 9 to 15 years calculated for the $SF_6$ time series at $20\,\text{hPa}$ and $40°\,\text{N}$

As described when discussing the polynomial coefficients, the $SF_6$ reference time series slightly bends in the 1980s which makes the lag time calculation around the late 1980s more sensitive to the fit interval length. The specific years of stronger variation in lag time shift slightly for different locations, dependent on the transport times to that location. As seen from Fig. 11 (right), 1983 to 1990 is also the time of strongest deviation for the non-suitable parameter selection in Fig. 11. The variation in lag time contributes to the variation in mean age in the beginning of the time series dependent on the selection of parameters in the derivation.

Now that the range for the polynomial coefficients $b$ and $c$ from the fit to the SF$_6$ time series is specifically known from Fig 12, the sensitivity to $\Gamma$ for the different $b$ and $c$ can additionally be considered. Based on Eq.7, for a fixed lagtime $\tau$, the mean AoA $\Gamma$ can be calculated for the range of $b$, $c$ and different ratio of moments. The results are shown in Fig. 14. Here, $\tau$ was chosen to be 4.5 years which is a reasonable value to apply if we consider the values for $\tau$ in Fig.13.

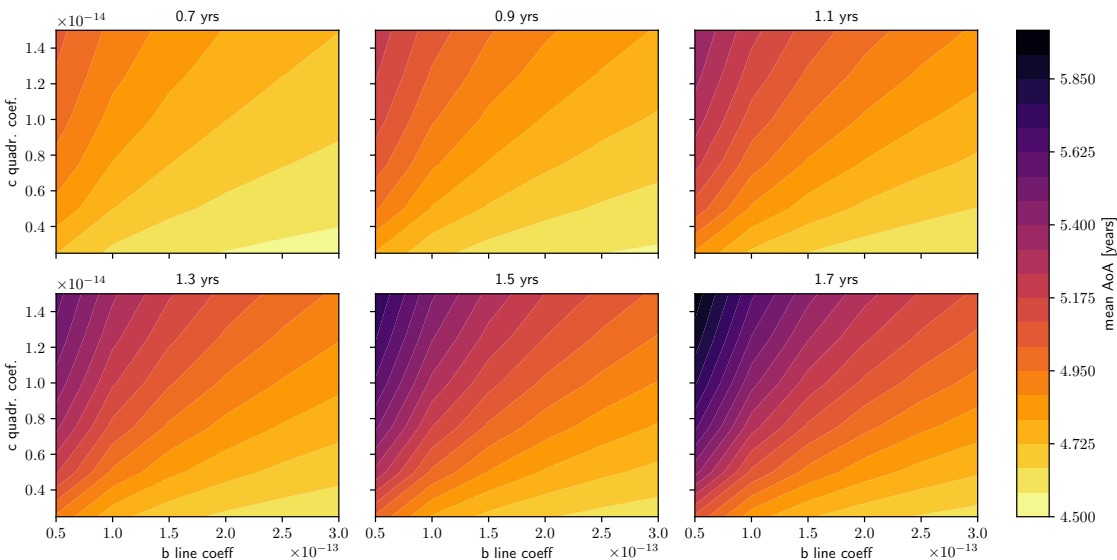

**Figure 14.** Mean AoA $\Gamma$ for the range of magnitude of the coefficients $b$ and $c$ based on Eq.7 using the fixed $\tau = 4.5$ years. The different panels show the respective ratio of moments (Headings).

5 It can be seen, that for $b$ close to zero, $\Gamma$ becomes more sensitive to the selection of the ratio of moments, as this range shows differences of up to one year in mean AoA, for the selected range of ratio of moments from 0.7 to 1.7 years. Most sensitivity of $\Gamma$ to the ratio of moment is given when in addition $c$ is on the larger end. If we consider the coefficients $b$ and $c$ in Fig. 12, around the year 1980, $b$ tends to be closer to zero and $c$ tends to be on the larger end. This means, that around 1980, the sensitivity of mean AoA to the coefficients is very pronounced. Further, the sensitivity is the way around, that smaller ratio of

10 moments give smaller mean AoA values. Smaller mean AoA values in the beginning of the time series lead to a less negative AoA trend.

To conclude the considerations why the AoA trend is sensitive to the parameter selection, it can be said, that due the shape of the SF$_6$ boundary condition the beginning of the AoA time series is particularly sensitive to the parameter selection. The bend in boundary condition around 1980 leads to sensitivity of the lag-time $\tau$. Further, the rather flat slope in the beginning

15 of the SF$_6$ time series implies more sensitivity of mean AoA to the parameter selection then. Therefore, the results about the sensitivity of mean AoA trends to the parameter selection are not dependent on our model or our model resolution.

The latter analyses were done for both $SF_6$ time series, the one prescribed in the simulation and the one used for calculations from observations. Only the results obtained from the $SF_6$ time series prescribed in the model are shown here as the results were similar.

## 3.3 AoA calculations via the Convolution Method

As described in Sec. 2.3.2, applying the convolution method weights the reference input according to the strength of its influence at a certain observation location. This avoids the influence of an assumption of the fraction of input, so one would expect to see more reliable agreement between AoA and its trends than when applying the fit method. Thus, the convolution method primarily requires an assumption about the ratio of moments of the assumed spectrum. The amount of years of reference can just be considered relatively high, e.g. 25 years. As a first example, Fig. 15 shows mean AoA and its trend calculated for ratio of moments 0.7 years and 1.25 years from the $SF_6$ tracer and from the linear tracer. One can see that for 1.25 years of ratio of moment, the agreement between AoA calculated from the linear tracer and from $SF_6$ becomes very good. However, for a ratio of moments of 0.7 years the disagreement prevails. Similar to the fit method, deviations in mean AoA are found in the beginning of the AoA time series maximizing in the 1980s, resulting in an AoA trend that is less negative.

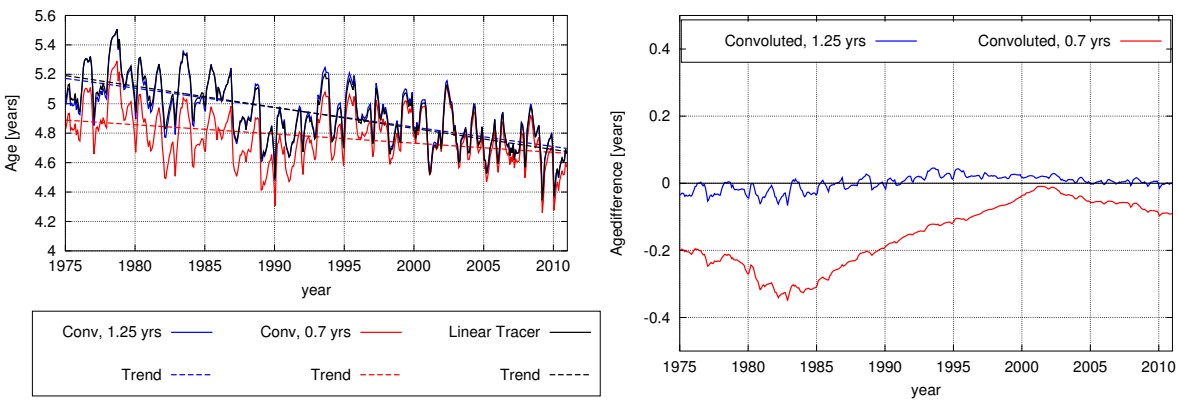

**Figure 15.** Left, mean AoA and its trend derived from $SF_6$ using the convolution method for 0.7 and 1.25 years ratio of moments (red and blue) and from the linear tracer (black) at 20 hPa and $40°$ N from 1975 to 2011 from the RC1-base-07 simulation. Right, the difference in mean AoA derived from $SF_6$ and the linear tracer.

Similar to Fig. 8, Fig. 16 shows, height-latitude plots of the mean absolute deviation between mean AoA calculated using the linear increasing tracer and the $SF_6$ tracer in color and the difference in their trends in $10^{-1}$ years decade$^{-1}$ in white contours. The location of the balloon borne observations by Engel et al. (2017) is indicated by the red boxes. Here one can see that overall, a ratio of moments of 0.7 years, which was applied to the balloon borne observations so far, leads to too small negative or even positive trends, which are insignificant. Specifically, in the range of the observation this is a deviation in mean AoA of 2.8% and in trend of 54%. This remains similar to applying this ratio of moments in the fit method. On the other hand, using the ratio of 1.25 years yields very good agreement in the range of the balloon borne observations (0.3% in mean AoA,

6% in trend). Again, using the spatial distribution of ratio of moments derived from the extended model spectra, finds good agreement, too (0.4% in mean AoA, 14% in AoA trend in the range of the observations).

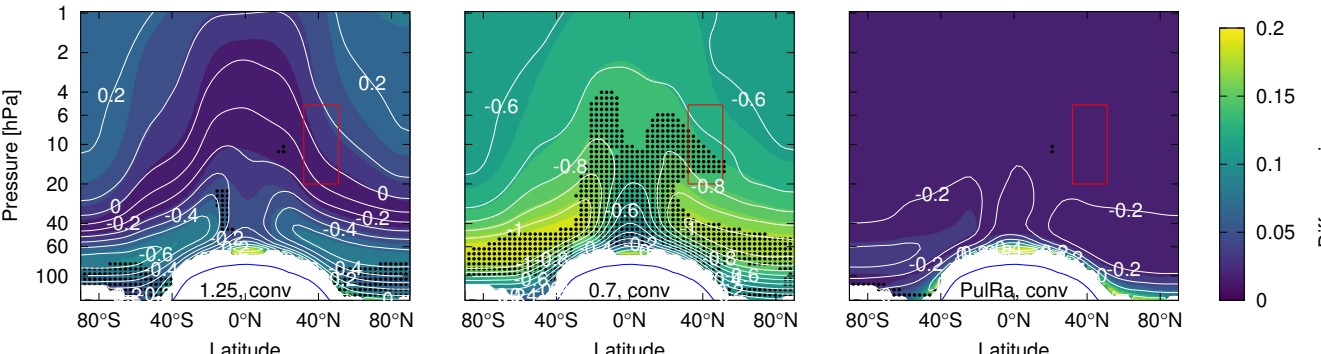

**Figure 16.** As Fig. 8, in color are the mean absolute deviation of mean AoA calculated from $SF_6$ from AoA derived from an ideal tracer for every month from 1975 to 2011 for different ratio of moments using the convolution method. The left panel is 1.25 years of ratio of moments and the middle is 0.7 years. PulRa denotes the ratio of moments derived from the tranPul simulation annual mean extended age spectra (right) The white contours are the difference in trends in $10^{-1}$ years decade$^{-1}$. White areas indicate where the derivation is not applicable. Black stippling shows insignificance of the $SF_6$ trend from zero on the 95% confidence level. The trend calculation considered the time series from 1975 to 2011. The red box shows the range of the balloon observations.

## 4 Discussion

### 4.1 Model Results

It has been shown here, that the variation of the mean AoA trend due to the selection of parameters is larger than the internal variability (e.g. Fig. 7). One might suggest that models still have spin-up effects as the deviations in mean AoA are often in the beginning of the time series. Both simulations, tranPul and RC1-base-07, were started in 1950, but the first ten years were considered as spin-up. So the data is used starting in 1960, such that the first mean AoA values are calculated for 1975, as the mean AoA calculation fits the reference time series backwards. Previous tests on the effects of spin-up time on AoA in EMAC

support that our mean AoA values should not be considerably influenced by spin-up effects for this amount of spin-up. Hence, the differences between models and the observations are not due to spin-up effects in models. We find that the major source of variation in $SF_6$-derived mean AoA especially using the fit method is the selection of parameters.

    Another question we consider is whether a second order fit is appropriate to describe the $SF_6$ reference time series. Testing higher order fits shows, that the 3rd order coefficients would be two orders of magnitude smaller than the second order coeffi-

cients and the higher ones even smaller. Therefore, we remain with the second order assumption. Especially since higher order calculations would require even more assumptions about the transport time distribution which would be even more difficult to constrain in measurement applications.

The convolution method enables to avoid uncertainties about the amount of time considered and the function that is fitted to the time series. Still, there is a high sensitivity to the ratio of moments and it remains open which ratio of moments is describing the atmosphere best. Still, it is rather certain, that larger ratios of moments are more realistic.

## 4.2 Recalculation of AoA from balloon borne observations

5   It is most valuable to put the results derived from the model data concerning parameters and methods of AoA calculations into context by testing whether we see the same tendencies in calculations of observed AoA. The mean age values are recalculated from the observations by Engel et al. (2017) with the fit method with different parameters and the convolution method with different ratio of moments. The results for the trends are given in Fig. 17 and Table 1.

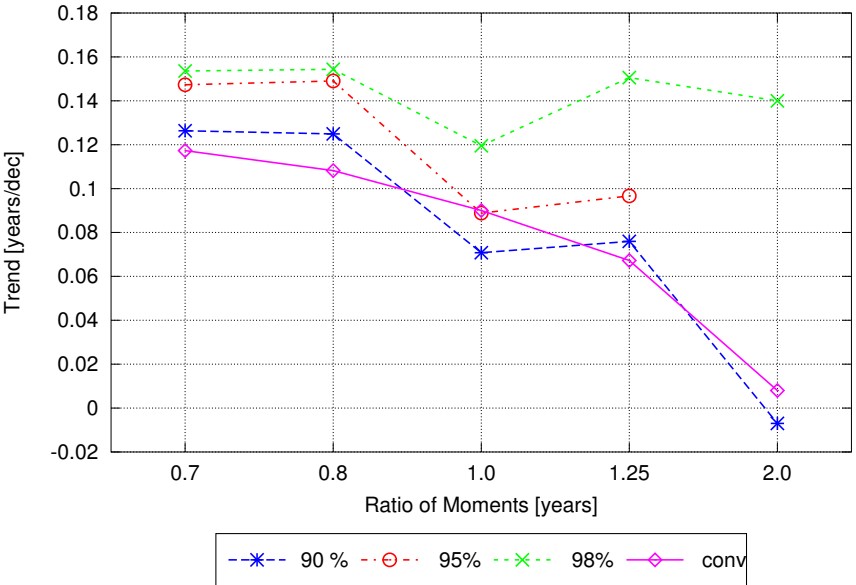

**Figure 17.** Trend in mean AoA derived from the observations for the fit and the convolution method for the respective fraction of input and ratio of moments.

**Table 1.** AoA trends recalculated from observations using fit method and convolution method for different parameters. Within the presented decimals, the errors are equal.

| Method | Trend [years decade$^{-1}$] calculated from ratio | | |
|---|---|---|---|
| | 0.7 years | 1.25 years | 2.0 years |
| Fit, 98% | 0.15 ($\pm$0.16) | 0.15 ($\pm$0.16) | 0.14 ($\pm$0.16) |
| Fit, 90% | 0.12 ($\pm$0.16) | 0.07 ($\pm$0.16) | $-$0.007 ($\pm$0.16) |
| Convolution | 0.11($\pm$0.16) | 0.07($\pm$0.16) | 0.008($\pm$0.16) |

In agreement to what we presented for the model data, there is little influence of the ratio of moments on the trend using the fit method for 98% fraction of input, as was used in the calculations so far. For 90% fraction of input, the trend becomes less positive and eventually even slightly negative for larger ratio of moments. Thus, we see the same tendencies towards smaller, respectively more negative trends for larger ratio of moments and smaller fraction of input.

Applying the convolution method for larger ratio of moments gives smaller positive trends, too. This confirms what we have seen from the model data. For both methods for the larger ratio of moments negative trends as seen in the models lay within the margins of error.

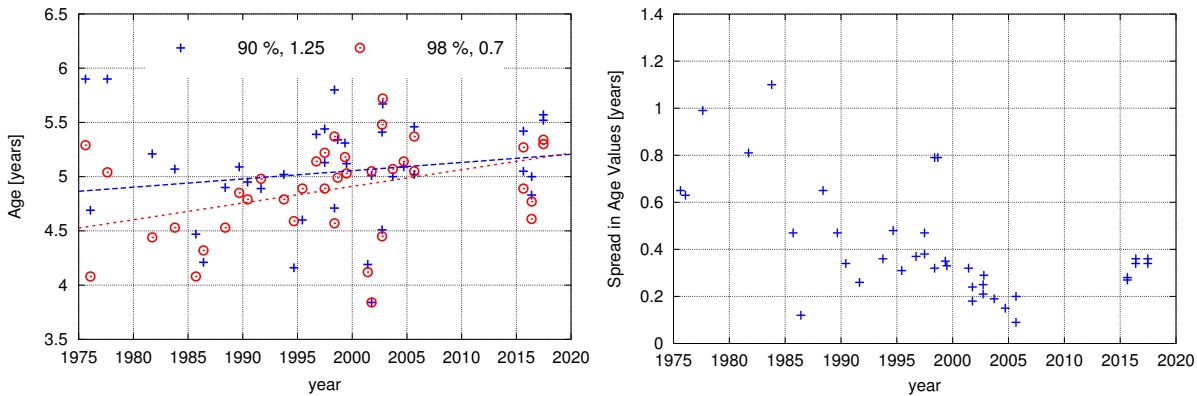

**Figure 18.** Left, example AoA time series derived from the observations using F=90%, 1.25 years ratio of moments and F=98%, 0.7 years. Right, the difference between maximum and minimum mean AoA found from the observations for the considered parameters (0.7, 0.8, 1. 0 and 1.25 years for fit and convolution method and 90%, 95% and 98% for the fit method).

Fig. 18 presents the AoA time series recalculated from the observations. On the left, the former parameter selection of 98% and 0.7 years for the fit method is displayed together with the recalculated version for 90% and 1.25 years. The right shows the
10 spread in mean AoA for the different parameter and methods, that is 0.7, 0.8, 1. 0 and 1.25 years for convolution method and 90% 95% and 98% for the fit method. The difference between the largest and smallest AoA value is shown. It becomes clear from the two plots, that the beginning of the $SF_6$ time series is more sensitive in the calculation, as seen for the model data.

Overall our results show that the selection of parameters helps resolve the difference between model and observation AoA trends. Furthermore, it should be mentioned that Volk et al. (1997) found good agreement with Waugh et al. (1997) using a
15 ratio of moments of 1.25 years for airborne in-situ measurements of $SF_6$. Applying this ratio of moments to the balloon borne observations results in mean AoA trends that are the same as those of the model, within the margins of error.

However, one should consider that the trend in Engel et al. (2017) was derived from both $SF_6$ and $CO_2$ measurements. For calculations using $CO_2$ measurements methane oxidation is considered as a minor source. In our discussion, we focused on the specific derivation methods themselves, such that we did not consider $CO_2$. Consequently, it was favorable to use a $SF_6$
tracer without mesospheric sinks. In particular, since the balloon borne observations we want to understand are located between 5-30 hPa and 32-51° N where the influence of the mesospheric $SF_6$ sinks has so far been considered as small. In addition, the

combination of $CO_2$ and $SF_6$ balloon borne observations would make an influence of $SF_6$ sinks on the mean AoA trend less clear. However, the $SF_6$ sinks are important to consider when discussing the specific differences between mean AoA derived from the global MIPAS satellite $SF_6$ observations and models as Linz et al. (2017) did, especially in the light of recent studies that report shorter lifetimes for $SF_6$ than assumed so far (Leedham Elvidge et al., 2018; Ray et al., 2017).

## 5   Conclusions

We investigated the sensitivity of the derivation of mean AoA to parameter choices that are necessary in the fit and convolution methods using a model as test-bed. References available in a model such as a linearly growing AoA tracer and age spectra allow specific testing of the methods.

In our analysis we were able to find a systematic variation of the mean AoA trend derived from $SF_6$ dependent on the ratio of moments and the fraction of input used in the AoA calculation when applying the fit method. The mean AoA values in the beginning of the time series are particularly sensitive to the ratio of moment and become larger with larger ratio of moments leading to more negative trends. In addition, the larger the fraction of input is, the more towards positive the trend. While the sensitivity to the ratio of moments was included in the error range by Engel et al. (2009), the usage of a fixed fraction of input of 98% reduced the sensitivity to the ratio of moments in their study.

Investigating another method to derive mean AoA, namely the convolution method, showed the same relation of smaller negative AoA trends for larger ratio of moments. The convolution method only assumes the ratio of moments. It provides very good agreement between AoA from $SF_6$ and the linear tracer for the correct ratio of moment.

Applied to the balloon borne observations was the fit method with a small ratio of moments (0.7 years) and a big fraction of input (98%), which both tend to shift the trend towards more positive values. Based on model results, we find better agreement with a linear AoA tracer for ratio of moments of about 1.25 and fraction of input of 95%. Therefore, the parameter selection helps to resolve why the mean AoA trends found in Engel et al. (2009) and Engel et al. (2017) are positive, while models show negative trends.

Our model results suggest using ratio of moments larger than 0.7 years. Using ratio of moments within the range 1.25-2 years is in line with what other 3d-model studies suggest (Waugh et al., 1997; Hauck et al., 2019). Specifically, using a spatial distribution is favorable as it considers aspects such as the broadening of the spectra due to mixing lower in the stratosphere.

Testing the impact of the parameter choice on AoA derived from the balloon borne observations, shows the same sensitivities for fit and convolution method as seen in the model. Within the margins of error the trend in AoA from observations and EMAC agree when assuming large ratio of moments (2 years). However, as the ratio of moments in the real atmosphere is currently unknown, we cannot conclude whether such a large ratio of moments is realistic. Overall, understanding the systematic influence of the parameter selection on the trend of mean AoA provides further insight in addition to the existing work on the uncertainties of mean AoA and its trends. This work clearly highlights the benefits of the consistent model evaluation of methods that are applied to observations.

*Data availability.* The ESCIMo data used in this study can be obtained through the British Atmospheric Data Centre (BADC) archive (ftp://ftp.ceda.ac.uk). For instructions for access to both archives see http://blogs.reading.ac.uk/ccmi/ badc-data-access. The tranPul data are available at Fritsch et al. (2020), https://doi.org/10.5281/zenodo.3826623. The observational balloon data and mean age values are available upon request. Due to the ongoing process of updating calibration scales, it is preferred that possible users contact the data providers directly

to discuss possible limitations and issues with the observations.

*Author contributions.* The initial concepts to derive mean AoA were initialized by HB and AE. HG and FF initiated the method evaluation. The simulation execution was done by RE with development contributions by FF. HB further contributed discussion and insight to the method evaluations. AE did the recalculation of the observational data. FF performed the data analysis and wrote the main parts of the paper. All authors contributed in finalizing the paper.

*Competing interests.* The authors declare that they have no conflict of interest.

*Acknowledgements.* This study was funded by the Helmholtz Association under grant VH-NG-1014 (Helmholtz-Hochschul-Nachwuchsforschergruppe MACClim). We also acknowledge the project ESCiMo (Earth System Chemistry integrated Modelling), within which the EMAC simulations were conducted at the German Climate Computing Centre DKRZ through support from the Bundesministerium für Bildung und Forschung (BMBF). Further, we want to thank both anonymous referees for their reviews. Thanks to Romy Heller for providing a review prior to

submitting.

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
