# Peer review of "Sensitivity of Age of Air Trends on the derivation method for non-linear increasing inert $SF_6$"

_Atmospheric Chemistry and Physics, 2019_

## Referee Comment (RC1) · Anonymous Referee #1 · 21 Dec 2019

This paper examines the uncertainties involved in estimating the mean age of stratospheric air (AoA) and its trend from observations of non linearly increasing tracers (here, mainly SF6 ). A large part of our knowledge of the stratospheric circulation comes from observational estimates of AoA. In this study, the authors evaluate the uncertainties of the AoA estimation method from measurements (Engel et al., 2009, 2017) by applying it to model SF6 data and comparing the obtained estimate to the actual model AoA trend. The paper's topic is of interest to ACP's readership and the analyses presented are thorough. The paper is definitely worth publishing. However, I believe the presentation could be made more pedagogic and would like the authors to carefully consider my comments below. For this reason I recommend major revisions.

[Figure]

**Main comments**

1) Presentation: The uncertainties discussed by the authors stem from the non linearity of SF6 increase at the boundary, which is illustrated in Fig.1. This non linearity is small in recent years but larger at earlier times when the time series is more convex (Fig. 10). As a consequence, the small sensitivity of the AoA retrieval on the chosen parameters for the 2010s becomes much larger for the 1980s when there is a kink in the time series, as illustrated in Fig. 9  12.

While this is clear in the manuscript, I think it should be stated earlier in the text. I would recommend the authors to first present the uncertainties in age estimates from SF6 at different times due to the shape of the curve (Fig. 9 to 12) before discussing the impact on the trend, and the improvements obtained by taking variable ratio of moments from the model (Fig. 5 to 8). Figure 10 in particular comes exclusively from the analysis of the time series and should be presented earlier on, to explain what might be expected.

-Implication of non-linearity of SF6 for the uncertainties in AoA estimates

-Quantification of uncertainties and improvements obtained by constraining the ratio of moments

2) In the abstract p1 line 22 the authors mention both SF6 and $CO_2$ observations being used to derive AoA, but in the model they only examine SF6. A nice addition would be to also discuss $CO_2$, but it might be beyond the scope of this study.

3) One of the messages I take from this paper is that mean age of air derived from observations may not be the optimal quantity to compare the model with. A better approach would be to introduce the tracers ($CO_2$, SF6) in the model and to compare the age estimates obtained applying the same method to modeled and observed tracers. The authors may want to put that forward in their discussion and evtl. abstract.

**Specific comments and typos**

title: "non-linearly increasing inert tracers"

p1 l 22: If possible, it would be interesting to quantify the uncertainty of the method for CO2.

P2 : in the introduction you first present the concept of age spectrum and then the mean age which is its first moment. I would do the same in Sect. 3 p3, i.e. put 2.1.1 and Eq. (2) first and then define mathematically the first moment of the spectrum and its relation to a linearly increasing tracer (Eq. 1. )

p2 l 27-31: In this paragraph you mention two sources of uncertainty for deriving AoA from measurements (non linearity and non inertness). As far as I understand you only address the non linearity, and your EMAC simulation does not have the SF6 meso-spheric sink. Is that correct? It should be stated here (and in Sect. 2.2 that your EMAC SF6 is inert).

P3 l 1: maybe replace "the non-linearity of the SF6 tracer boundary condition" ?

P3 Sect. 3 : See above: I would first introduce the age spectrum and its relation to the boundary condition of an inert surface-emitted tracer. Then introduce mean AoA.

P3 l20: the reference location is the surface here, correct?

p4 l1 :"quadraic" → Quadratic

P4 Eq. (3): please define t' and t0

p4 Eq (4): reference for this equation (Volk et al., 1997 ?)

p4 line 11: please provide a formula to define gamma and delta mathematically. Also note that delta**2 is not the second moment strictly speaking (there is a factor of 1/2 involved here)

p4 lines 13-14 : "approximating the reference time series by a second order function": a second order polynomial (defined in Eq. 1).

p4 Eq. 5 : Introduce a symbol, e.g. $F$ for the fraction of input

p4 line 24: Convolution Method "Numerical convolution method". The other method also implies an analytical convolution with the fit polynomial

p 5 line 2-3: "this approach requires only an assumption about the ratio of moments": this is not true, there is an additional assumption: that the spectrum is an inverse Gaussian. Although the fit method also uses that assumption to estimate $t_{fit}$ for a given fraction of input $F$, once the quadratic fit is chosen Eq (4) is exact whatever the shape of the age distribution might be. Actually, you might say that the numerical convolution and the quadratic fit both use an analytical approximation, but to different functions: one approximates the time series by a second order polynomial, the other approximates the spectrum by an inverse Gaussian. The uncertainty related to the fit method is arguably easier to constrain. Anyway, combining the two methods might provide a better idea about the AoA uncertainty, as you show in the discussion.

P5 : How long are the simulations? Are you confident that the early part of the simulation is not affected by spin-up? I would already raise that issue here.

P6 l2-3 : well-constrained: What do you mean exactly? Is it just that time series shifted?

P6 line 4: interpolated : extrapolated ?

P6 line 6: "released at the surface": globally?

P7 l1: "it is fairly easy": "it is straightforward (Eq. 1)"

p7 l3: How different is the mean AoA between RC1... and transpulse?

P7 l 10 : "spectra resolution" : "transit time resolution"

p8: legend of fig. 3: are you doing a fit or of eq 6 ? Or using the width and mean age

estimated from the spectrum?

P8 line 12 :"we can expect" : "we expect"

p9 l3: "selection": "section"

p10 line 1: "it is clearly evident" : "it is evident"

p 11: Fig. 6 the different subfigures need to be explained in the legend

p 11 l 8: $F = 98$

p12 Fig. 7: I agree this biennal and 11 year cycle in the ratio is interesting and deserves further investigation

p12 Fig. 8 : This is the main figure of the paper,

p12 legend of Figure 8: which confidence test are you using ?

P13 l6: "In such cases a reference location closer to the troposphere should be used.": What do you mean?

P14, Figure 10: it would be interesting to see the time series together with exemplary fits

p14 l 20: As I mentioned above, I am not at all convinced that the numerical convolution method is better

p16 Figure 13 and p11 Figure 6: Is the mean AoA displayed here a time average? Over what period?

P16 line 16: spin-up: I think this information should be given earlier, in the model description.

P17 l 8-9: please consider my comment above regarding the advantages of the convolution method. Again, I prefer the term "numerical convolution".

P17 l 10: 'it is rather certain, that larger ratio of moments are more realistic." : You

should specify according to the model (Figure 4).

p19 line 19: "the larger the ratio of moments...": I would state here again that the retrieved AoA is older in the 1980s with larger ratio of moments while it is left unchanged in the 2010s, so that larger AoA imply more negative trends

p19 line 30: "the parameter selection helps to resolve ...": Please rephrase, for instance: "Hence, the discrepancy between model results and the Engel et al; observations may be partly explained by the choice of parameters for the retrieval in Engel et al."

As far as I understand, the discrepancy is not completely resolved since the recalculation in section 4.2 still doesn't agree with the model. This should also be clearly stated.

p20: One of the messages I get from your study is that, to evaluate transport in models and compare them to observations, it would be preferable and more direct to implement SF6 like tracers (rather than using estimated and modeled AoA). Would you agree? This could be put in the conclusion.

---

## Referee Comment (RC2) · Anonymous Referee #2 · 10 Jan 2020

Review of "Sensitivity of Age of Air Trends on the derivation method for non-linear increasing tracers"

In this study, the authors examine how specific choices for method of converting SF6 concentrations to Age of Air (AoA) impact the resulting value and trend. They use output from the EMAC model, which includes both a linear age tracer and a tracer whose boundary condition matches the observed timeseries of SF6. They evaluate two methods for calculating AoA: In one method the width of the age spectrum is varied, along with th e"fraction of the age spectrum to be considered". In the other method, the "ratio of moments" is varied. The different methods lead to different results, and these results may serve to illuminate the consistent apparent positive trend in midlatitude AoA obtained from previous SF6 balloon measurements (Engel et al 2009, 2017).

Overall this is a good study. The experiments seem well thought out and the analysis is thorough. The result that the trend in age could be this biased by the method for calculating it is extremely important. In fact, this result could be highlighted a bit more. I believe that this will be suitable for publication in ACP following major revisions to improve the introduction and the theory sections, in particular. With these improvements in explanation of context and methods, I believe the manuscript will be much more impactful, and I am excited to cite it in my own work.
See below.

There are a few major points where significant additional discussion would be helpful.
1) A more extended discussion of the methods is needed.
    a. The ratio of moments, for example, is not defined mathematically or explained physically
    b. Although the "fraction of input" is defined, it's not clear what it means. Having never done the calculation this way myself (like the majority of your readers, probably), my intuition for this particular quantity is lacking.
    c. Appropriate citations for 2.1 and 2.1.1 should be included. Hall and Plumb 1994 or Waugh and Hall 2002?. Equation 6 and the discussion around it should be moved to 2.1.1 and this discussion expanded to include the possibility of other shapes such as two peaks (e.g. Andrews et al. 2001). Then the Li et al and Ploeger and Birner references make sense, as they are expanding the discussion by referring to time-varying age spectra.
    d. Why does your convolution method not involve iteration? E.g. Stiller et al. 2012 section 3.3. They find no major difference between using w=0.7 and w=1.5, and this may be due to their iterative process. Since this is one of the main methods that has been used to calculate age from SF6 observations, it should be included here.

       A diagram (or multiple diagrams) may be helpful in expanding this theory section. Show what the difference between w=0.7 and w=1.5 looks like, show what is meant by "fraction of input", etc. Since the crux of this paper is modifying these variables, it seems essential that the reader have a clear understanding of what they are.

2) Some discussion of why the larger ratio of moments causes a lower (more negative) trend seems important. This should be demonstrable analytically, I think. But at least describe the qualitative argument for why this should be the sense of the difference. Adding this will avoid any concern about these results being model dependent or method dependent.

3) The context/introduction for this paper is missing some literature and is a bit incorrect in a few places.

    a. Paragraph 1: Hall and Plumb (1994) does not actually discuss age in quantifying the BDC except with respect to being able to predict transport of other tracers. No vertical velocities appear in that paper. The quantitative relationships between AoA and the circulation were really first established in pressure coordinates using the tropical leaky pipe model in Neu and Plumb (1999). Ray et al. (2016) improved upon this framework, incorporating other tracer information as well. Linz et al. (2016) also built upon Neu and Plumb and isolated the diabatic circulation as a function of the horizontal difference in age of air between the tropics and the extratropics. In a follow up paper, they quantitatively calculated the overturning strength from tracer observations to compare with a model (Linz et al. 2017).

    b. Paragraph 2: There is not really a discrepancy between the apparent increasing trend in age of air observations and an increasing trend in the circulation strength itself. There are a number of reasons for this:

        i. The age of air at one latitude and height is not directly related to the BDC. It is a function of the overturning and of the mixing (e.g., Garny et al. 2014). It is the difference in age between the tropics and extratropics that is quantitatively related to the BDC, specifically the diabatic circulation (Linz et al. 2016, Li et al. 2018).

        ii. Meanwhile, model trend calculations have all been done for the residual circulation in pressure coordinates (e.g. Butchart et al. 2010). The atmospheric circulation itself is expanding (Singh and O'Gorman 2012), which has the result of "accelerating" the stratospheric circulation if one only looks in pressure coordinates (Oberlander-Hayn et al. 2016).

        iii. The observed trend in age found from the MIPAS data in Haenel et al. 2015 is not uniformly positive. Also, that length of record (10 years) is insufficient to determine a real trend in the BDC, due to the internal variability (Hardiman et al. 2017).

        iv. The work by Garcia et al. (2011) demonstrated that with the sparse sampling of the balloon measurements in Engel et al. 2009, a positive trend could easily be found in an overall decreasing timeseries (Fig. 12).

        Although it hasn't been done, between Garcia's results and Hardiman's results, I suspect the additional AirCore points of Engel et al. 2017 would still not be sufficient to properly sample the forced trend.

    c. Paragraph 3: Specify that you are looking at SF6 measurements. There are plenty of other issues that apply to determining age from other tracers, and this study has enough in it without getting into CO2 or non-clock tracers. You probably want to change the title of the paper to reflect this. Perhaps title something like "Are

positive trends in stratospheric age of air from SF6 measurements an artifact of the nonlinearity of the SF6 timeseries?" Hall and Waugh 1998 should be mentioned here also for context.

    d. Paragraph 4: Missing relevant literature on SF6 lifetime: associative electron attachment has newly been shown to be dominant beneath 105 km (Totterdill et al. 2015, Kovacs et al. 2017)

Detailed comments:

p.2

l.13 cause→case

l.20 more closely

p.3 l.7 and many other places "allows to": allow takes an object before its verb. So "allows *somebody* to *do something*" is the proper construction. Alternatively, "allows use of… and thereby better understanding of" would work here, since "use" and "understanding" are both objects.

p. 5 l 31 mention here that your SF6 doesn't have a mesospheric sink, so that aspect of uncertainty in AoA calculation is not included in your examination. (a strength of this study, I believe)

p. 6 l 4 Completely interpolated from what?

Fig. 3 and discussion thereof. Why exactly is the resolution only 3 months? I assume this is related to your pulse setup, so perhaps explain it explicitly there.

p.8 l.3 Can you be slightly more explicit about this extension—specifically, how is "second half" defined?

p.8 l.6—does this mean that the inverse Gaussian form is problematic? If so, state that.

Discussion of Figure 6 is confusing. State explicitly what was done, why, and what the results show.

p. 11 l.2 Could this be because of your 9.5 year tracer reset? Unless this actually lines up with the solar cycle well, in which case, show that.

Fig. 8 caption: just repeat the description here for the middle and right panels rather than making the reader go back to Fig. 6.

Discussion of Figs. 10 and 11: I got lost here. Could you add a summary the conclusions from these two after describing them?

p. 19 l. 5 "the trend" what trend? Also, this is not a good topic sentence for the rest of the paragraph. You don't need to talk about methane oxidation, unless there's a specific point to make here.

p. 19 l. 12 Linz et al. 2017 did this, i.e. compared MIPAS SF6 to model(s?) while accounting for the sink, and the sink was very limiting.

p. 20. End with a stronger statement than this! Your work is much more important than your concluding paragraph implies. Something about needing to understand much more about details of age spectra before calculating forced trends? Your study is another excellent example of why there is not actually a clear disagreement between measurements and models—the measurements are both too limited and too limiting to calculate a trend the same way that is possible for the models.

Figures:
Single panel figures have large enough labels, but the labels on multipanel figures are very small. Fig. 7 also has small labels. Label locations for colorbars are not consistent.

Fig. 3: choose another color (or line style) for one of MAM or JJA, as this is not currently colorblind friendly. Could just switch Hall and Plumb color with MAM.

Fig 10 Just one power of 10, probably 14 for both panels

References:
Neu and Plumb, 1999 JGR "Age of air in a "leaky pipe" model of stratospheric transport"
Ray et al. 2016 doi: 10.1002/2015JD024447
Linz et al. 2016 doi:10.1175/JAS-D-16-0125.1
Linz et al. 2017 doi: 10.1038/NGEO3013
Li et al. 2018 doi: 10.1002/2017JD027562
Butchart et al. 2010 doi: 10.1175/2010JCLI3404.1
Singh and O'Gorman 2012 doi: 10.1175/jcli-d-11-00699.1
Oberlander-Hayn et al. 2016 doi: 10.1002/2015GL067545
Hardiman et al. 2017 doi: 10.1002/2017GL072706
Hall and Waugh 1998 JGR "Influence of nonlocal chemistry on tracer distributions: Inferring the mean age of air from SF6"
Totterdill et al. 2015 doi: 10.1021/jp5123344
Kovacs et al. 2017 doi: 10.5194/acp-17-883-2017

---

## Author Comment (AC1) · 21 Feb 2020

Thanks to Perter Haynes for guiding through the editorial process. We would like to thank both referees for their helpful review.

The very useful comments lead to improvements of the manuscript. The biggest changes are found in the Introduction and the Methods section. The Introduction is now extended and more accurate concerning the scientific background. The Methodology is now introduced in more detail and clearly and was supplemented by sketches to facilitate a better understanding of what was done. Other sections were adjusted by smaller changes.

Detailed answers to the issues raised by the Referees are given below. First we will consider issues, that both referees raised comments about and that are beneficial to consider in context to each other. Afterwards, comments by Referee #1 and Referee #2 that can be considered separately will be answered. The manuscript version with the marked-up differences (latexdiff) is attached at the end of this document.

**1   Answers to both Referees**

**1.1   Title**

**Minor Comment Referee #1**

*title: "non-linearly increasing inert tracers"*

**Major Comment Referee #2**

*3) c. ... Specify that you are looking at SF6 measurements. There are plenty of other issues that apply to determining age from other tracers, and this study has enough in it without getting into CO2 or non-clock tracers. You probably want to change the title of the paper to reflect this. Perhaps title something like "Are positive trends in stratospheric age of air from SF6 measurements an artifact of the nonlinearity of the SF6 timeseries?" ...*

**Reply:** The phrasing of the title was considered and adjusted to:
"Sensitivity of Age of Air Trends on the derivation method for non-linear increasing inert $SF_6$"

**1.2   Concerns about the Method part**

**Main Comment Referee #2**

*1)  A more extended discussion of the methods is needed.*

    *a.  The ratio of moments, for example, is not defined mathematically or explained physically*

    *b.  Although the "fraction of input" is defined, it's not clear what it means. Having never done the calculation this way myself (like the majority of your readers, probably), my intuition for this particular quantity is lacking.*

    *c.  Appropriate citations for 2.1 and 2.1.1 should be included. Hall and Plumb 1994 or Waugh and Hall 2002?. Equation 6 and the discussion around it should be moved to 2.1.1 and this discussion expanded to include the possibility of other shapes such as two peaks (e.g. Andrews et al. 2001). Then the Li et al and Ploeger and Birner references make sense, as they are expanding the discussion by referring to time-varying age spectra.*

    *d.  Why does your convolution method not involve iteration? E.g. Stiller et al. 2012 section 3.3. They find no major difference between using w=0.7 and w=1.5, and this may be due to their iterative process. Since this is one of the main methods that has been used to calculate age from SF6 observations, it should be included here. A diagram (or multiple diagrams) may be helpful in expanding this theory section. Show what the difference between w=0.7 and w=1.5 looks like, show what is meant by "fraction of input", etc. Since the crux of this paper is modifying these variables, it seems essential that the reader have a clear understanding of what they are.*

**Minor Comment Referee #1**

- *P2 : in the introduction you first present the concept of age spectrum and then the mean age which is its first moment. I would do the same in Sect. 3 p3, i.e. put 2.1.1 and Eq. (2) first and then define mathematically the first moment of the spectrum and its relation to a linearly increasing tracer (Eq. 1. )*

- *P3 Sect. 3 : See above: I would first introduce the age spectrum and its relation to the boundary condition of an inert surface-emitted tracer. Then introduce mean AoA.*

- *P3 l20: the reference location is the surface here, correct?*

- *p4 l1 :"quadraic" → Quadratic*

- *P4 Eq. (3): please define t' and t0*

- *p4 Eq (4): reference for this equation (Volk et al., 1997 ?)*

- *p4 line 11: please provide a formula to define gamma and delta mathematically. Also note that delta\*\*2 is not the second moment strictly speaking (there is a factor of 1/2 involved here)*

- *p4 lines 13-14 : "approximating the reference time series by a second order function": a second order polynomial (defined in Eq. 1).*

- *p4 Eq. 5 : Introduce a symbol, e.g. F for the fraction of input*

- *p4 line 24: Convolution Method "Numerical convolution method". The other method also implies an analytical convolution with the fit polynomial*

- *p 5 line 2-3: "this approach requires only an assumption about the ratio of moments": this is not true, there is an additional assumption: that the spectrum is an inverse Gaussian. Although the fit method also uses that assumption to estimate $t_{fit}$ for a given fraction of input F, once the quadratic fit is chosen Eq (4) is exact whatever the shape of the age distribution might be. Actually, you might say that the numerical convolution and the quadratic fit both use an analytical approximation, but to different functions: one approximates the time series by a second order polynomial, the other approximates the spectrum by an inverse Gaussian. The uncertainty related to the fit method is arguably easier to constrain. Anyway, combining the two methods might provide a better idea about the AoA uncertainty, as you show in the discussion.*

**Reply:** Thanks to both referees for the very informative helpful comments on making the Method section of the paper more understandable. The Method section was restructured such that it starts with a now more detailed explanation based on the age spectrum and then provides the different derivation methods. Further, two schematic plots were added. The missing formula definitions were included. The required citations were added. As the changes were quite extensive, they are best understood from the manuscript version with the marked-up differences.

**1.3   SF$_6$ sinks**

**Minor Comment Referee #1**

*p2 l 27-31: In this paragraph you mention two sources of uncertainty for deriving AoA from measurements (non linearity and non inertness). As far as I understand you only address the non linearity, and your EMAC simulation does not have the SF$_6$ mesospheric sink. Is that correct? It should be stated here (and in Sect. 2.2 that your EMAC SF$_6$ is inert).*

**Minor Comment Referee #2**

*p. 5 l 31 mention here that your SF6 doesn't have a mesospheric sink, so that aspect of uncertainty in AoA calculation is not included in your examination. (a strength of this study, I believe).*

**Reply:** In both cases suggested by Referee #1 we added the information that our $SF_6$ tracer is inert. Therefore, we did not further include it in the section about the convolution method as Referee #2 suggested it, as the inertness is relevant to fit method and convolution method.

**1.4 $SF_6$ timeseries**

**Minor Comment Referee #1**

*P6 line 4: interpolated : extrapolated?*

**Minor Comment Referee #2**

*p. 6 l 4 Completely interpolated from what?*

**Reply:** The phrasing was corrected.
"Before that, $SF_6$ is estimated from emissions. Prior to 1961, the data is linearly extrapolated."

**1.5 Conclusions**

**Minor Comment Referee #1**

*p19 line 30: "the parameter selection helps to resolve ...": Please rephrase, for instance: "Hence, the discrepancy between model results and the Engel et al; observations may be partly explained by the choice of parameters for the retrieval in Engel et al." As far as I understand, the discrepancy is not completely resolved since the recalculation in section 4.2 still doesn't agree with the model. This should also be clearly stated.*

**Minor Comment Referee #2**

*p. 20. End with a stronger statement than this! Your work is much more important than your concluding paragraph implies. Something about needing to understand much more about details of age spectra before calculating forced trends? Your study is another excellent example of why there is not actually a clear disagreement between measurements and models -the measurements are both too limited and too limiting to calculate a trend the same way that is possible for the models.*
**Reply:** Well, those comments are a bit opposing.

Considering Referee #1, we specifically phrased, that both trends agree within the margins of error (i.e. $1\sigma$). The margins of error that were estimated for the observations by Engel et al. 2017 are simply that large as also shown in the table of recalculations. Before, EMAC AoA trends and the observation were more than $2\sigma$ apart.

The only addition made to the conclusion is the closing sentence, which now reads:
"This work clearly highlights the benefits of the consistent model evaluation of methods that are applied to observations."

**2 Answer to Anonymous Referee #1**

**2.1 Major Comments**

**Major Comment 1**

*1) Presentation: The uncertainties discussed by the authors stem from the non linearity of SF6 increase at the boundary, which is illustrated in Fig.1. This non linearity is small in recent years but larger at earlier times when the time series is more convex (Fig. 10). As a consequence, the small sensitivity of the AoA retrieval on the chosen parameters for the 2010s becomes much larger for the 1980s when there is a kink in the time series, as illustrated in Fig. 9 12. While this is clear in the manuscript, I think it should be stated earlier in the text. I would recommend the authors to first present the uncertainties in age estimates from SF6 at different times due to the shape of the curve (Fig. 9 to 12) before discussing the impact on the trend, and the improvements obtained by taking variable ratio of moments from the model (Fig. 5 to 8). Figure 10 in particular comes exclusively from the analysis of the time series and should be presented earlier on, to explain what might be expected.*

- *Implication of non-linearity of SF6 for the uncertainties in AoA estimates*

- *Quantification of uncertainties and improvements obtained by constraining the ratio of moments*

**Reply:** We understand the idea of first explaining the specific uncertainties and then present the effects on the AoA trends. Though the order we chose, stills seems favorable to us. Generally, the uncertainties of both methods to derive mean AoA from non-linear tracers have already been discussed very carefully by Engel et al. (2009) and Leedham Elvidge et al. (2018), as is mentioned in the introduction. Therefore, for the reader at a first glance it might not seem necessary to investigate the time dependency of the uncertainty of deriving AoA. Rather, it might be beneficial, to first present the systematic impact on AoA trends as it has implications for ongoing discussions first.

**Major Comment 2**

*2) In the abstract p1 line 22 the authors mention both SF6 and CO2 observations being used to derive AoA, but in the model they only examine SF6. A nice addition would be to also discuss CO2, but it might be beyond the scope of this study.*
*Also minor comment: p1 l 22: If possible, it would be interesting to quantify the uncertainty of the method for $CO_2$.*

**Reply:** Indeed, in our study we want to clearly investigate the methods to derive mean AoA and its trends. Already the sinks of $SF_6$ were omitted to achieve a clear understanding of the systematic uncertainties of the methods to derive mean AoA. If $CO_2$ was to be included, further effects, namely the methane oxidation and the seasonal cycle, would need to be considered. This would blur the understanding of the systematic uncertainties of the methods. This was also put forward at the end of section 4.2 of the paper.

"However, one should consider that the trend in Engel et al. (2017) was derived from both $SF_6$ and $CO_2$ measurements. For calculations using $CO_2$ measurements methane oxidation is considered as a minor source. In our discussion, we focused on the specific derivation methods themselves, such that we did not consider $CO_2$. Consequently, it was favorable to use a $SF_6$ tracer without mesospheric sinks."
Omitting discussing $CO_2$ as a tracer is also supported by Referee #2.

*3) c. Paragraph 3: Specify that you are looking at $SF_6$ measurements. There are plenty of other issues that apply to determining age from other tracers, and this study has enough in it without getting into $CO_2$ or non-clock tracers.*

**Major Comment 3**

*One of the messages I take from this paper is that mean age of air derived from observations may not be the optimal quantity to compare the model with. A better approach would be to introduce the tracers (CO2, SF6) in the model and to compare the age estimates obtained applying the same method to modeled and observed tracers. The authors may want to put that forward in their discussion and evtl. Abstract.*

*Also minor comment: p20: One of the messages I get from your study is that, to evaluate transport in models and compare them to observations, it would be preferable and more direct to implement SF6 like tracers (rather than using estimated and modeled AoA). Would you agree? This could be put in the conclusion.*

**Reply:** This is definitely an interesting perspective on the results. Though, our conclusion would still be, to use all information available in the model. From its physical meaning, AoA from a linear tracer in the model is understood. To achieve the same physical meaning when deriving AoA from a non-linear tracer, the correct parameters still need to be applied. Such that the task of constraining the free parameters still persists. Even when deciding to compare AoA derived consistently in models and observations from non-linear tracers the free parameters can cause problems. The "correct parameter" is even expected to be different due to the respective biases, as e.g. differences between the actual mixing and the mixing representation in the model are expected to lead to different ratio of moments. The new,additional closing sentence takes the comment into consideration. "This work clearly highlights the benefits of the consistent model evaluation of methods that are applied to observations."

**2.2 Minor Comments**

- *P5 : How long are the simulations? Are you confident that the early part of the simulation is not affected by spin-up? I would already raise that issue here.*

- *P16 line 16: spin-up: I think this information should be given earlier, in the model description.*

**Reply:** A master student in our group did extensive testing on the effect of spin up on AoA variability. Therefore, we are indeed confident, that spin-up is not influencing our AoA results. To inform the reader earlier, we have added: "The simulation had ten years of spin-up, i.e. 1950-1959. This means, that for the earliest AoA values calculated for 1975 the respective fit back along the reference time series does not does not stretch into spin-up. Due to previous testing, we are confident, that the amount of spin-up is sufficient to no longer affect the mean AoA."

*P6 l2-3 : well-constrained: What do you mean exactly? Is it just that time series shifted?*
**Reply:** The $SF_6$ measurements at one location, i.e. at Cape Grim, are used to estimate an equatorial reference time series that can be linked to the amount entering the stratosphere.

*P6 line 6: "released at the surface": globally?*
**Reply:** Yes, globally. We specified this in the text which now reads: "In the tranPul simulation, periodical pulses of an inert tracer were released at the surface in the range between $20°$ S and $20°$ N."

*p7 l3: How different is the mean AoA between RC1 and transpulse?*
**Reply:** We did not recognize strong differences. The transpulse simulation was only used to estimate a spatial distribution of ratio of moments. Therefore, unless there were strong deviations, the specific AoA differences are not relevant here.

*p8: legend of fig. 3: are you doing a fit or of eq 6? Or using the width and mean age estimated from the spectrum?*
**Reply:** We did the latter. To make it more clear, the caption now reads: "The solid yellow line is the spectrum calculated from the Hall and Plumb parameterization using the values from the integration of annual mean spectrum, that is a mean age value of 1.6 years (vertical line) and a ratio of moments of 1.8 years (Eq. 2 and 3)"

*p9 l3: "selection": "section"*
**Reply:** Selection was meant, therefore we kept it.

*p12 legend of Figure 8: which confidence test are you using?*
**Reply:** A t-Test was used. To specify that, the caption was changed to:

"White areas are the areas that can not be analyzed for a trend and black stippling shows insignificance of the trend from zero on the 95% confidence level using a t-test."

*P13 l6: "In such cases a reference location closer to the troposphere should be used.": What do you mean?*

**Reply:** To clarify, the sentence was changed to:

"In such cases a reference location closer to the tropopause should be used to avoid artifacts of this tracer initialization in the AoA calculation."

– *p14 l 20: As I mentioned above, I am not at all convinced that the numerical convolution method is better*

– *P17 l 8-9: please consider my comment above regarding the advantages of the convolution method. Again, I prefer the term "numerical convolution".*

**Reply:** The term numerical convolution has been adapted when introducing the method. With the revised method section it might now be more clear, that the fit method also uses the inverse Gaussian assumption to determine the fit interval.

Overall, from a model perspective, specifying one method as better is not needed. Predominantly, we describe what we see in our results where we consider both methods, as you said. The consistently deviating trends using the fit method with a fraction of input of 98% illustrate quite well the difficulty of constraining two parameters when using the fit method. Maybe presenting an anecdote here illustrates that. Initially, when testing the fit method in EMAC, we only tried 98% fraction of input and different ratio of moments. As we always saw the same strong deviation from AoA from the linear tracer, we assumed there was an error in our calculation that we were looking for for quite some time, though there was none.

Further, even assuming that the assumption about the inverse Gaussian describes transport in the atmosphere very roughly, considering it weights the impact of the reference time series more correctly than putting the same weight on all points of the reference series. The issue of overly considering very old air when calculating mean AoA using the fit method are quite visible in our results.

*P14, Figure 10: it would be interesting to see the time series together with exemplary fits*

**Reply:** We made plots of this, though the differences in the polynomial coefficients are not as visible so that such a plot is not providing additional information. An example is provided in Fig. 1

*p16 Figure 13 and p11 Figure 6: Is the mean AoA displayed here a time average? Over what period?*

**Reply:** For every month considered, i.e. 1975 til 2011, the absolute deviation of mean AoA from $SF_6$ and the linear tracer was taken. Show is in the figure is the time average of that value. To make it more clear, the caption is now adjusted to:

"As Fig. 8, in color are the mean absolute deviation of mean AoA calculated from SF6 from AoA derived from an ideal tracer for every month from 1975 to 2011 for different ratio of moments using the convolution method. "

*p19 line 19: "the larger the ratio of moments...": I would state here again that the retrieved AoA is older in the 1980s with larger ratio of moments while it is left unchanged in the 2010s, so that larger AoA imply more negative trends*

**Reply:** Specifications were added.

**Reply:** The following simple comments were adapted or did not require any changes.

– *P3 l 1: maybe replace "the non-linearity of the SF6 tracer boundary condition" ?*

– *P7 l1: "it is fairly easy": "it is straightforward (Eq. 1)"*

– *P7 l 10 : "spectra resolution" : "transit time resolution"*

– *P8 line 12 :"we can expect" : "we expect"*

– *p10 line 1: "it is clearly evident" : "it is evident"*

[Figure]

**Figure 1.** Example of the SF$_6$ time series with a typical second order polynomial fit. In addition, the linear tracer is shown.

- *p 11: Fig. 6 the different subfigures need to be explained in the legend*

- *p 11 l 8: F=98*

- *p12 Fig. 7: I agree this biennal and 11 year cycle in the ratio is interesting and deserves further investigation*

- *p12 Fig. 8 : This is the main figure of the paper,*

- *P17 l 10: "it is rather certain, that larger ratio of moments are more realistic." : You should specify according to the model (Figure 4).*

**3 Answer to Anonymous Referee #2**

**3.1 Major Comments**

**Major Comment 1**

Most of this comment was already considered in the section discussing comments from both Referees. Here, the suggestion d. about Stiller et al. (2012) shall be given a more detailed answer. Though termed "iterative" they use a very similar method and rather embed it in a retrieval method to already have a first guess for $\Gamma$. That they do not not see a strong difference in AoA is rather attributed to the considered time span, as they only had the years 2002-2010 available. Even Haenel et al. (2015) only had the years 2002-2012 available. Due to the shape of the SF$_6$ time series, we do not see a strong dependency to the ratio of moments around that time either. The sensitivity of the mean AoA to the ratio of moments is predominantly present from 1975 to ca. 1985.

Further, an addition to remark 1) b. iv. shall be made. Though Garcia et al., 2011 considered the effect of the sampling of AoA data on the derived trends, the transfer to the Engel et al. 2017 (respectively Engel et al. 2009) trends is limited as Garcia et al. 2011 employed a different mean Aoa derivation approach. Consequently, the work of Garcia et al. has already been considered specifically later in the introduction in more detail.

**Major Comment 2**

*2) Some discussion of why the larger ratio of moments causes a lower (more negative) trend seems important. This should be demonstrable analytically, I think. But at least describe the qualitative argument for why this should be the sense of the difference. Adding this will avoid any concern about these results being model dependent or method dependent.*

5  *Also minor comment: Discussion of Figs. 10 and 11: I got lost here. Could you add a summary the conclusions from these two after describing them?*

**Reply:** To perform an actual analytic solution the convolution integrals for two different ratios of moments must be compared, which could not be solved analytically. Instead comparing the formula for mean AoA based on the fit method for two different ratios of moments still leaves the lagtime and the polynomial coefficients. Therefore, to consider the suggestion, we added Fig.

10  14 at the end of section 3.2, which shows the range of resulting mean AoA for different ratios of moments depending on the polynomial coefficients for a fixed $\tau$. The plot further illustrates how the beginning of the AoA time series is sensitive to the ratio of moments and therefore leads to sensitive trends. The end of the section then considers the additional minor comment. Now it reads as follows (Figure numbering incorrect):

"Now that the range for the polynomial coefficients $b$ and $c$ from the fit to the $SF_6$ time series is specifically known from Fig 12,

15  the sensitivity to $\Gamma$ for the different $b$ and $c$ can additionally be considered. Based on Eq.7, for a fixed lagtime $\tau$, the mean AoA $\Gamma$ can be calculated for the range of $b$, $c$ and different ratio of moments. The results are shown in Fig. 14. Here, $\tau$ was picked to be 4.5 years which is a reasonable value to apply considering Fig.13. It can be seen, that for $b$ close to zero, $\Gamma$ becomes

[Figure]

**Figure 2.** Results for mean AoA $\Gamma$ for the range of magnitude of the coefficients $a$ and $b$ based on Eq.**??** using the fixed $\tau = 4.5$ years. The different panels show the respective ratio of moments (Headings).

more sensitive to the selection of the ratio of moments, as this range shows differences of up to one year in mean AoA, for the selected range of ratio of moments from 0.7 to 1.7 years. Most sensitivity of $\Gamma$ to the ratio of moment is given when in

20  addition $c$ is on the larger end. Now considering Fig. 12, around 1980, $b$ tends to be closer to zero and c tends to be on the larger end. This means, that around the year 1980, the sensitivity of mean AoA to the coefficients is very pronounced. Further, the sensitivity is the way around, that larger ratio of moments give smaller mean AoA values. Smaller mean AoA values in the beginning of the time series lead to a less negative AoA trend.

To conclude the considerations why the AoA trend is sensitive to the parameter selection, it can be said, that due the shape of

25  the $SF_6$ boundary condition the beginning of the AoA time series is particular sensitive to the parameter selection. The bend in boundary condition around 1980 leads to sensitivity of the lag-time $\tau$. Further, the rather flat slope in the beginning of the $SF_6$ time series implies more sensitivity of mean AoA to the parameter selection then. Therefore, the results about the sensitivity of mean AoA trends to the parameter selection are not dependent on our model or our model resolution.

The latter analyses were done for both SF6 time series, the one prescribed in the simulation and the one used for calculations from observations. Only the results obtained from the SF6 time series prescribed in the model are shown here as the results were similar."

**Major Comment 3**

*3) The context/introduction for this paper is missing some literature and is a bit incorrect in a few places.*

    a. *Paragraph 1: Hall and Plumb (1994) does not actually discuss age in quantifying the BDC except with respect to being able to predict transport of other tracers. No vertical velocities appear in that paper. The quantitative relationships between AoA and the circulation were really first established in pressure coordinates using the tropical leaky pipe model in Neu and Plumb (1999). Ray et al. (2016) improved upon this framework, incorporating other tracer information as well. Linz et al. (2016) also built upon Neu and Plumb and isolated the diabatic circulation as a function of the horizontal difference in age of air between the tropics and the extratropics. In a follow up paper, they quantitatively calculated the overturning strength from tracer observations to compare with a model (Linz et al. 2017).*

    b. *Paragraph 2: There is not really a discrepancy between the apparent increasing trend in age of air observations and an increasing trend in the circulation strength itself. There are a number of reasons for this:*

        i. *The age of air at one latitude and height is not directly related to the BDC. It is a function of the overturning and of the mixing (e.g., Garny et al. 2014). It is the difference in age between the tropics and extratropics that is quantitatively related to the BDC, specifically the diabatic circulation (Linz et al. 2016, Li et al. 2018).*

        ii. *Meanwhile, model trend calculations have all been done for the residual circulation in pressure coordinates (e.g. Butchart et al. 2010). The atmospheric circulation itself is expanding (Singh and O'Gorman 2012), which has the result of "accelerating" the stratospheric circulation if one only looks in pressure coordinates (Oberlander-Hayn et al. 2016).*

        iii. *The observed trend in age found from the MIPAS data in Haenel et al. 2015 is not uniformly positive. Also, that length of record (10 years) is insufficient to determine a real trend in the BDC, due to the internal variability (Hardiman et al. 2017). iv. The work by Garcia et al. (2011) demonstrated that with the sparse sampling of the balloon measurements in Engel et al. 2009, a positive trend could easily be found in an overall decreasing timeseries (Fig. 12). Although it hasn't been done, between Garcia's results and Hardiman's results, I suspect the additional AirCore points of Engel et al. 2017 would still not be sufficient to properly sample the forced trend.*

    c. *Paragraph 3: Specify that you are looking at SF6 measurements. There are plenty of other issues that apply to determining age from other tracers, and this study has enough in it without getting into CO2 or non-clock tracers. You probably want to change the title of the paper to reflect this. Perhaps title something like "Are positive trends in stratospheric age of air from SF6 measurements an artifact of the nonlinearity of the SF6 timeseries?" Hall and Waugh 1998 should be mentioned here also for context. d. Paragraph 4: Missing relevant literature on SF6 lifetime: associative electron attachment has newly been shown to be dominant beneath 105 km (Totterdill et al. 2015, Kovacs et al. 2017)*

**Reply:** Thank you, Referee #2 for this very informative and well understandable feedback. The suggested additions to the scientific background were considered in the revised introduction of the manuscript. The full information about the changes in the revised manuscript can be found in the manuscript version with the marked-up differences .

**3.2 Minor Comments**

*Fig. 3 and discussion thereof. Why exactly is the resolution only 3 months? I assume this is related to your pulse setup, so perhaps explain it explicitly there.*

**Reply:** This is now specified to in the beginning of the paragraph.

"The implementation was such, that tracer pulses were released every three months and kept running for 9.5 years, which is due to the technical realization."

*p.8 l.3 Can you be slightly more explicit about this extension-specifically, how is "second half" defined?*

**Reply:** This is now described more specifically:

"Following Ploeger et al. (2016), the spectra were extended to 50 years using an exponential fit on the second half of the spectrum, i.e. transit times above 5 years."

*p.8 l.6-does this mean that the inverse Gaussian form is problematic? If so, state that.*

**Reply:** No, it rather means that the assumed range of 0.7 to 1.25 years of ratio of moments for the inverse Gaussian is to small. Hall and Plumb (1994) initially only considered 10 years of age spectrum in their approximation. This is why their calculations yield a smaller ratio of moments than other studies with extended spectra. To make it more clear, the following sentence was

10   added.

"Though, it should be considered that **?** assumed 10 years of age spectrum in their approximation, while the other approaches also considered longer transit times, which tends to give larger ratio of moments."

*Discussion of Figure 6 is confusing. State explicitly what was done, why, and what the results show.*

15   **Reply:** The beginning of the paragraph has been rearranged to make it more clear. It now reads (Additional Figures changed the Figure numbering):

"To built onto what was already seen from Fig. 7, Fig. 8 shows the mean absolute deviation of mean AoA derived from $SF_6$ and the linear tracer in the latitude-height domain for different selections of parameters in the derivation (colors). The white contours show the difference in trends in $10^{-1}$ years decade$^{-1}$. Further, in Fig. 8 we also consider a spatial distribution of

20   ratio of moments. Even though we do not have age spectra available for the RC1-base-07 simulation, those from the tranPul simulation are a suitable first estimate for EMAC as a model. We considered the ratio of moments for the 9.5 year long spectra (PulRa sh) and the extended spectra (PulRa) (see Fig. 6)."

*p. 11 l.2 Could this be because of your 9.5 year tracer reset? Unless this actually lines up with the solar cycle well, in which*

25   *case, show that.*

**Reply:** The spectra are transient, i.e. pulses are continuously released every three months. Therefore they cannot cause such an artifact. We assumed the solar cycle as an impact of that cycle length. However, as the temporal variation of the ratio of moments did not pose a relevant impact on our calculations, we deemed the investigation of the temporal variation as out of the scope of the paper.

*p. 19 l. 5 "the trend" what trend? Also, this is not a good topic sentence for the rest of the paragraph. You don't need to talk about methane oxidation, unless there's a specific point to make here.*

**Reply:** The trend was specified (See below). Methane oxidation is important for the consideration of $CO_2$ as a second tracer besides $SF_6$ in our analysis. As Referee #1 suggested to also consider $CO_2$, we feel obliged to specify the requirements for that.

35   As this paragraph also contains the link between modeled and observed AoA, the reader may come up with these questions here. We therefore feel we should also discuss it here.

"However, one should consider that the trend in Engel et al. (2017) was derived from both $SF_6$ and $CO_2$ measurements."

*p. 19 l. 12 Linz et al. 2017 did this, i.e. compared MIPAS SF6 to model(s?) while accounting for the sink, and the sink was*

40   *very limiting.*

**Reply:** Very good point, thanks. Now it is mentioned as:

"However, the SF6 sinks are important to consider when discussing the specific differences between mean AoA derived from the global MIPAS satellite SF6 observations and models as Linz et al. (2017) did, especially in the light of recent studies that report shorter lifetimes for SF6 than assumed so far (Leedham Elvidge et al., 2018; Ray et al., 2017)."

45

*Fig. 7 also has small labels. Label locations for colorbars are not consistent.*

**Reply:** Unfortunately, we are not exactly sure, what was meant concerning the colorbars. Hopefully, it has resolved with adaptions made to improve captions and such.

**Reply:** The following simple comments were adapted.

- *p.2 l.13 cause → case*

- *p.2 l.20 more closely*

- *p.3 l.7 and many other places "allows to": allow takes an object before its verb. So "allows \*somebody\* to \*do something\*" is the proper construction. Alternatively, "allows use of... and thereby better understanding of" would work here, since "use" and "understanding" are both objects.*

- *Fig. 8 caption: just repeat the description here for the middle and right panels rather than making the reader go back to Fig. 6.*

- *Single panel figures have large enough labels, but the labels on multipanel figures are very small.*

- *Fig. 3: choose another color (or line style) for one of MAM or JJA, as this is not currently colorblind friendly. Could just switch Hall and Plumb color with MAM.*

- *Fig 10 Just one power of 10, probably 14 for both panels*

[revised manuscript text omitted]

---

## Referee Report (RR1)

The authors have addressed my concerns. This is an excellent and important paper, and I look forward to citing it in my own work.

I appreciate the additional explanation of methods, and the example plots that are now included.
The legend in Fig. 18 is quite hard to read.

---

## Referee Report (RR2)

Review of Fritsch et al., ACPD, 2019

The authors have taken the referees' comments into account and restructured the presentation of the methodology, which improved the paper. I still have a few requests (listed below). They mainly concern the analytical formulas and their consistency with the figures. After those points have been clarified, the paper should be suitable for publication.

- I noticed that Eq.7 (formerly Eq. 3) has been changed between the two versions. It is important that the (co-)authors double check for typos since varying the parameters of that equation is the crux of the paper. I would rather get (-b/c)*tau for the last term under the square root (with ½ removed). Moreover, there is also a sign ambiguity in the current Eq. 7 . The authors should explain which sign is picked under which conditions. (I think that the minus is preferred in most cases, if not all).

- The signs of the coefficients shown in Fig. 12 seem inconsistent with the sign convention in Eq. (6), given the shape of the curve in Fig.3 . In particular, I would expect b to be always positive (since the linear trend of SF6 corresponds to an increase of SF6 with increasing time). Please check. Furthermore, Eq. (7) shows that the relevant parameter for the retrieved age is the ratio b/c, and not the specific value of either b or c. The authors might consider plotting the ratio in Fig. 12 as well.

- Again, the authors might consider showing Fig. 14 using b/c.

---

## Author Response (AR2)

Once again many thanks to the editor and both referees. Due to the already very helpful previous reviews, we only required to make small changes to the manuscript, which will be explained in the following.

**1 Answer to Referee #1**

*The authors have addressed my concerns. This is an excellent and important paper, and I look forward to citing it in my own work.*
*I appreciate the additional explanation of methods, and the example plots that are now included.*
*The legend in Fig. 18 is quite hard to read.*
**Reply:** Thank you. The legend of Figure 18 was slightly enlarged to improve readability.

**2 Answer to Referee #2**

- *I noticed that Eq.7 (formerly Eq. 3) has been changed between the two versions. It is important that the (co-)authors double check for typos since varying the parameters of that equation is the crux of the paper. I would rather get (-b/c)\*tau for the last term under the square root (with 1/2 removed). Moreover, there is also a sign ambiguity in the current Eq. 7. The authors should explain which sign is picked under which conditions. (I think that the minus is preferred in most cases, if not all).*

- *The signs of the coefficients shown in Fig. 12 seem inconsistent with the sign convention in Eq. (6), given the shape of the curve in Fig.3 . In particular, I would expect $b$ to be always positive (since the linear trend of $SF_6$ corresponds to an increase of $SF_6$ with increasing time). Please check. Furthermore, Eq. (7) shows that the relevant parameter for the retrieved age is the ratio b/c, and not the specific value of either $b$ or $c$. The authors might consider plotting the ratio in Fig. 12 as well.*

- *Again, the authors might consider showing Fig. 14 using b/c.*

**Reply:** The comment aided in improving the clarification. As you pointed out correctly, the factor 2 in the last term of of the square root of Eq. 7 was indeed misprinted, though it is correct in our analysis script.
In addition, the definition of the linear coefficient $b$ was subject to some inconsistency. That is due to the respective time axis used. Our analysis script implementation of the reference time series was done using a positive transit time axis (i.e. ranging from 0 to e.g. 15 years). This relates to the nominal year axis now termed $\tilde{t}$ which ranges from 1960 to 2011 by the reference time $t_0$ (which might have a value of e.g. 1975)

$$t = -(\tilde{t} - t_0) \tag{1}$$

Now, in the case of Eq. 6 which was meant to be expressed by the nominal year time axis this would result in

$$\chi_0(\tilde{t}) = a + b(\tilde{t} - t_0) + c(\tilde{t} - t_0)^2 = a - bt + t^2 \tag{2}$$

Which leads to the switch in sign of the linear term, depending on which way around the polynomial is defined. As the representation along the the nominal year axis is maybe more intuitive considering Fig. 3, the representation was unified accordingly which affects Fig. 12 (left) and 14. This probably clarifies the signs in Eq. 7 which have to be picked consistently. Still, as apart from the mixed up labels of Fig. 12 and 14 and Eq. 6, our analysis implementation was consistent, no result actually changed. To also make this clear in the manuscript, it now reads:
"The fit performed can be expressed either relative to the nominal time axis $\tilde{t}$ ranging from 1960 to 2011 and the reference time $t_0$ or by the transit time $t$ (note the change in sign):

$$\chi_0(\tilde{t}) = a + b(\tilde{t} - t_0) + c(\tilde{t} - t_0)^2 = a - bt + t^2 \tag{3}$$

"

Further, thanks to the comment, a missing minus sign in Eq. 4 was corrected.

Next, we want to answer your comment showing the ratio of the polynomial coefficients $b/c$ instead or in addition to the coefficients $b$ and $c$ in Fig. 14. As is clear from Fig. 12 (right), the denominator $c$ changes sign around the years 1995-2000. Which means, that the ratio $b/c$ shows divergent behavior then, that leads to this representation being not as insightful. Therefore, we have not included it is this discussion.

[revised manuscript text omitted]

---

## Author Response (AR3)

Dear editor,

Please find attached the required files for typesetting and production. In addition to the accepted version, the reference to uploaded data was added.

Regards,

Frauke Fritsch